# Multiscale assessment of North American terrestrial carbon balance

**Kelsey T. Foster**[1,2], **Wu Sun**[1], **Yoichi P. Shiga**[3], **Jiafu Mao**[4], and **Anna M. Michalak**[1,2]

[1]Department of Global Ecology, Carnegie Institution for Science, Stanford, 94305 TS1, USA
[2]Department of Earth System Science, Stanford University, Stanford, 94305 TS2, USA
[3]independent researcher: San Francisco, CA, USA
[4]Environmental Sciences Division and Climate Change Science Institute,
Oak Ridge National Laboratory, Oak Ridge, 37830 TS3, USA

**Correspondence:** Kelsey T. Foster (ktfoster@stanford.edu)

**Abstract.** Comparisons of carbon uptake estimates from bottom-up terrestrial biosphere models (TBMs) to top-down atmospheric inversions help assess how well we understand carbon dioxide ($CO_2$) exchange between the atmosphere and terrestrial biosphere. Previous comparisons have shown varying levels of agreement between bottom-up and top-down approaches, but they have almost exclusively focused on large, aggregated scales (e.g., global or continental), providing limited insights into reasons for the mismatches. Here we explore how consistency, defined as the spread in net ecosystem exchange (NEE) estimates within an ensemble of TBMs or inversions, varies with at finer spatial scales ranging from $1° \times 1°$ to the continent of North America. We also evaluate how well consistency informs accuracy in overall NEE estimates by filtering models based on their agreement with the variability, magnitude, and seasonality in observed atmospheric $CO_2$ drawdowns or enhancements. We find that TBMs produce more consistent estimates of NEE for most regions and at most scales relative to inversions. Filtering models using atmospheric $CO_2$ metrics causes ensemble spread to decrease substantially for TBMs, but not for inversions. This suggests that ensemble spread is likely not a reliable measure of the uncertainty associated with the North American carbon balance at any spatial scale. Promisingly, applying atmospheric $CO_2$ metrics leads to a set of models with converging flux estimates across TBMs and inversions. Overall, we show that multiscale assessment of the agreement between bottom-up and top-down NEE estimates, aided by regional-scale observational constraints is a promising path towards identifying fine-scale sources of uncertainty and improving both ensemble consistency and accuracy. These findings help refine our understanding of biospheric carbon balance, particularly at scales relevant for informing regional carbon-climate feedbacks.

## 1 Introduction

Reliable estimates of carbon dioxide ($CO_2$) uptake by the terrestrial biosphere are necessary for understanding both historical and future climate change because the terrestrial biosphere mitigates anthropogenic $CO_2$ emissions by storing carbon in above- and below-ground biomass. Net ecosystem exchange (NEE) of $CO_2$ cannot be measured directly at scales greater than a plot ($\sim 1\,km^2$); that is, the scale of the footprint of an eddy covariance flux tower (Kljun et al., 2015). Estimates at broader scales therefore rely either on "bottom up" methods such as terrestrial biosphere models (TBMs) that represent process-based understanding of flux drivers (e.g., Hayes et al., 2012; Sitch et al., 2015) or on "top down" methods such as atmospheric inverse models that attribute observed atmospheric variability in $CO_2$ concentrations to upwind biospheric activity (e.g., Ciais et al., 2011; Gourdji et al., 2012; Gurney et al., 2002; Michalak et al., 2004; Peylin et al., 2013; Shiga et al., 2018a; Thompson et al., 2016).

Understanding of net biospheric carbon uptake has improved through comparisons between TBMs and inversions (Hayes et al., 2012; King et al., 2015), but discrepancies both between these approaches and between specific models using either approach have persisted even in relatively well studied regions such as North America and Europe. The re-

sulting uncertainties in carbon flux estimates limit our ability to anticipate carbon–climate feedbacks, and therefore to assess the impacts of alternative emission pathways and related climate mitigation policies (Friedlingstein et al., 2014 TS4; Huntzinger et al., 2017; King et al., 2015). Uncertainties both between modeling approaches and between specific models arise from the specific characteristics of TBMs and inversions.

For TBMs, model performance is limited by incomplete understanding of underlying processes (Hayes et al., 2012; Huntzinger et al., 2012, 2017; Schwalm et al., 2010, 2019; Seiler et al., 2022). One major source of uncertainty is that models may incorporate different key mechanisms or represent them with varying levels of detail. For example, nitrogen limitation likely reduces $CO_2$ fertilization, but coupled carbon–nitrogen dynamics are not included in all models (Bonan and Levis, 2010; Brovkin and Goll, 2015; Jain et al., 2009; Sokolov et al., 2008; Tharammal et al., 2019; Thornton et al., 2009; Wieder et al., 2015), and permafrost thaw, which can cause release of carbon from high latitudes (Burke et al., 2013; Koven et al., 2013), is not mechanistically resolved in most models. Large discrepancies also arise from different approaches to modeling land use change, vegetation dynamics, and fire (Ahlström et al., 2015; Bastos et al., 2020; Hardouin et al., 2022; Tharammal et al., 2019). Beyond such structural uncertainties, models parameterize processes differently and often use different driving data, which introduces additional uncertainty (Huntzinger et al., 2017; Jung et al., 2007; Lovenduski and Bonan, 2017; Schwalm et al., 2010, 2019). Previous studies have addressed the use of different driver data by establishing model ensembles that adhere to a standardized protocol and utilize consistent forcing data, but doing so does not fully account for observed uncertainties (Huntzinger et al., 2013, 2020; Friedlingstein et al., 2020; Sitch et al., 2015). Understanding and addressing these key sources of uncertainty in TBMs helps to improve our understanding of biospheric carbon exchange

For inversions, uncertainties arise from the limited information content of available atmospheric observations and from choices made in the statistical setup of the model. More specifically, the choice of prior estimates, prior error correlations, observational data, transport model, boundary conditions, data assimilation time period, and model resolution all lead to differences across models (Ciais et al., 2010; Göckede et al., 2010; Kondo et al., 2020; Michalak et al., 2017; Peylin et al., 2013). Paradoxically, even regions with relatively high data availability, such as Europe, can still exhibit large spread in carbon uptake estimates derived from inverse models (Kondo et al., 2020; Monteil et al., 2020). This suggests that other aspects of the inverse modeling framework may be more significant contributors to uncertainty, though there is not a clear consensus on the main source of uncertainty (Gaubert et al., 2019; Kondo et al., 2020). For example, some studies have found transport errors to be a primary source of uncertainty (Peylin et al., 2011; Schuh et al.,

2019), while other studies have shown that fossil fuel emission uncertainties play a key role in variability (Gurney et al., 2005; Peylin et al., 2011; Saeki and Patra, 2017). Another limitation is that atmospheric $CO_2$ observations, and inversions based on these observations, do not directly inform process-level understanding of the controls on carbon uptake to the extent that bottom-up approaches can (Baker et al., 2006; Gurney et al., 2004).

Given the uncertainties inherent to both approaches and their complementary strengths, comparisons between ensembles of TBMs and inversions can be particularly helpful in diagnosing our understanding of the carbon balance of the terrestrial biosphere. If TBMs and inversions yield similar estimates, this agreement increases confidence in the reliability of both model types as sources of realistic information. Previous studies, however, have come to different conclusions about the degree of agreement between bottom-up and top-down estimates (Bastos et al., 2020; Canadell et al., 2011; Kondo et al., 2020; Hayes et al., 2018). Some comparisons have shown agreement between NEE estimates from TBMs and inversions (Ciais et al., 2010; King et al., 2015; Sitch et al., 2015), while others have shown a large amount of variability both within and between bottom-up and top-down ensembles (Bastos et al., 2020; Chevallier et al., 2014; Huntzinger et al., 2012, 2017; Schwalm et al., 2019; Sun et al., 2021). For instance, King et al. (2015) found bottom-up and top-down methods agree on the sign of the North American land sink, albeit with notable spread in estimates, while Kondo et al., found bottom-up and top-down approaches disagree in many countries at the regional scale. Furthermore, Chevallier et al. (2014) showed that an ensemble of inversions had significant disagreement at the hemispheric and regional scales and Huntzinger et al. (2012) reported a lack of consensus among TBMs in determining whether North American land functions as a carbon sink or source.

A common approach for comparing TBMs and inversions is to examine the agreement across the mean of model ensembles (Ciais et al., 2010; Hayes et al., 2012; King et al., 2015). Here we use the term "agreement" between estimates to refer to the degree to which NEE estimates from an ensemble of TBMs differ from NEE estimates from an ensemble of inversions. In addition, analyzing the variability or spread amongst TBMs or inversions provides insights into the degree of consistency in carbon flux estimates. Here we use the term "consistency" of estimates to refer to the degree of variability in NEE estimates within an ensemble of either TBMs or of inversions.

Previous comparisons between bottom-up and top-down estimates have revealed that the agreement between estimates depends on the spatial scale and the region. At the global scale, inversions yield more consistent NEE estimates than TBMs (Friedlingstein et al., 2022 TS5), largely due to the global constraint provided by atmospheric $CO_2$ observations. However, at regional scales, consistency is limited for both TBMs and inversions and it is even difficult to de-

termine whether certain regions are a net sink or source of $CO_2$ (Ciais et al., 2013; Kondo et al., 2020) due to the uncertainties associated with both approaches outlined above. For North America, agreement between bottom-up and top-down estimates has improved over time, but this apparent agreement is in part due to the large range of estimates (i.e., low consistency) for both TBMs and inversions (Hayes et al., 2012; King et al., 2012, 2015; Pacala et al., 2001). This scale- and region-dependent agreement makes it difficult to determine the optimal path towards reducing uncertainties. This challenge is in part because comparisons of bottom-up and top-down methods are primarily conducted at large aggregated scales – for example, global, hemispheric, and continental scales (Bastos et al., 2020; Ciais et al., 2010; Hayes et al., 2012; Huntzinger et al., 2012; Pacala et al., 2007), which aids little in attributing causes of observed mismatches (Bastos et al., 2020; Hayes et al., 2012; Kondo et al., 2020).

A key step forward is to look at agreement beyond large, aggregated scales. Looking across multiple spatial scales provides a more in-depth understanding of the level of agreement between carbon budgets from bottom-up and top-down approaches. Despite this, few studies have taken this approach, especially for sub-continental spatial scales. Agreement between bottom-up and top-down estimates at global and hemispheric scales (Bastos et al., 2020; Kondo et al., 2020; Sitch et al., 2015) is a necessary but not sufficient condition for reconciling differences in carbon budgets at regional scales (Kondo et al., 2020). When large spread in model estimates makes it difficult to determine whether large regions such as Europe, boreal Asia, Africa, South Asia, and Oceania are even net sinks or sources (Kondo et al., 2020), multiscale comparisons may shed new light on the lack of consistency. Gourdji et al. (2012) compared bottom-up and top-down models at sub-continental scales and found better agreement during the growing seasons than in the dormant season, allowing for a more in-depth and focused exploration into the reasons for the observed (dis)agreement. The key insights gained from multiscale comparisons highlight the need for a more comprehensive comparison of bottom-up and top-down NEE estimates across spatial scales (Gourdji et al., 2012; Kondo et al., 2020).

Examining the agreement between bottom-up and top-down methods across spatial scales can also provide insights into the relationship between consistency, agreement, and accuracy in model predictions. Assessing agreement between bottom-up and top-down budget estimates is not necessarily equivalent to determining the accuracy of carbon budgets, however (Knutti et al., 2010; Kondo et al., 2020; Lovenduski and Bonan, 2017). Instead, accuracy should be assessed against observational constraints. There have been efforts, such as the International Land Model Benchmarking Project (ILAMB), to evaluate model accuracy by quantifying agreement between reference datasets and model outputs across multiple statistical metrics (Collier et al., 2018). Model skill scores are useful in assessing agreement between reference data and model data, but it is possible to misinterpret model performance without careful analysis of the metrics that make up the overall skill score (Bonan et al., 2019; Collier et al., 2018). In addition, reference data to which models are compared are often themselves modeled data products (e.g., FLUXCOM is used as reference data for gross primary productivity (GPP); Seiler et al., 2022). Model evaluation against atmospheric $CO_2$ observations can in principle provide more direct insights into the variability and accuracy of model NEE estimates (Fang et al., 2014; Fang and Michalak, 2015; Sun et al., 2021). By comparing carbon budget estimates to atmospheric $CO_2$ observations, in addition to comparing these estimates across spatial scales, we can determine the degree to which consistency within an ensemble is representative of accuracy. While better model performance under current conditions does not necessarily indicate better performance under future conditions, this approach helps with model improvement by allowing for quick identification of which models yield more realistic results. Subsequently, in-depth analysis can be done to identify key model characteristics that lead to improved results.

Here, we compare large ensembles of bottom-up and top-down model estimates of NEE for North American across various spatial scales to assess how consistency in model estimates varies across scales and between modeling approaches (Table 1). We expect inversions to be more consistent at larger scales thanks to the constraint provided by atmospheric observations, and TBMs to be more consistent at smaller spatial scales because they are informed by process-based understanding. We then evaluate whether greater consistency corresponds to higher accuracy and lower uncertainty in overall NEE estimates. We determine if the consistency within ensembles and the agreement between top-down and bottom-up approaches are impacted by ensemble subsetting; that is, limiting ensembles to models that can reproduce basic aspects of the variability, magnitude, and seasonality of atmospheric $CO_2$ observations. We expect consistency to improve when ensembles include only models that agree with basic features of atmosphere $CO_2$ observations, thereby also increasing the degree to which consistency informs accuracy, or, in other words, making model spread a more apt measure of uncertainty.

## 2 Data and methods

### 2.1 Data

#### 2.1.1 Model ensembles

Estimates of NEE from three model ensembles were used (Table 1). Bottom-up estimates came from two TBM intercomparison projects, namely, the Multi-scale Synthesis and Terrestrial Model Intercomparison Project (MsTMIP-v2; Huntzinger et al., 2013, 2018, 2020; Wei et al., 2014a,

**Table 1.** Names and references for the models included in each ensemble used in this study. The models that meet the variability, seasonality, magnitude, or all three metrics are indicated by an "X". An asterisk indicates the models that are present in both MsTMIP-v2 and TRENDY-v9 ensembles (CLM BG1, ISAM BG1, and VISIT SG3), in this case the metrics, were evaluated based on TRENDY-v9 versions (see Sect. 2.1.1), but the MsTMIP-v2 versions were used when evaluating the impact of how NEE is defined on consistency and agreement (Sects. 3.1 and 3.3.3, TS6 Fig. S7).

| Ensemble | Model name | Reference | Variability $N_{TBMs} = 9$ $N_{Inversions} = 8$ | Magnitude $N_{TBMs} = 4$ $N_{Inversions} = 6$ | Seasonality $N_{TBMs} = 10$ $N_{Inversions} = 6$ | All three metrics $N_{TBMs} = 3$ $N_{Inversions} = 6$ |
|---|---|---|---|---|---|---|
| MsTMIP-v2 (BG1) | BIOME-BGC | Thornton et al. (2002) | | | X | |
| MsTMIP-v2 (BG1) | CLASS-CTEM-N+ | Huang et al. (2011) | | | | |
| MsTMIP-v2 (BG1) | CLM4VIC | Lei et al. (2014) | | | | |
| MsTMIP-v2 (BG1) | DLEM | Tian et al. (2011) | | | | |
| MsTMIP-v2 (SG3) | GTEC | Ricciuto et al. (2011) | X | | X | |
| MsTMIP-v2 (BG1) | JPL-CENTURY | Parton et al. (1988) | | | | |
| MsTMIP-v2 (SG3) | JPL-HYLAND | Levy et al. (2004) | | | | |
| MsTMIP-v2 (SG3) | LPJ-wsl | Sitch et al. (2003) | | | | |
| MsTMIP-v2 (SG3) | ORCHIDEE-LSCE | Krinner et al. (2005) | | | X | |
| MsTMIP-v2 (SG3) | SiB3 | Baker et al. (2008) | X | | | |
| MsTMIP-v2 (SG3) | SiBCASA | Schaefer et al. (2008) | X | X | X | X |
| MsTMIP-v2 (BG1) | TEM6 | McGuire et al. (2010) | X | X | | |
| MsTMIP-v2 (BG1) | TRIPLEX-GHG | Zhu et al. (2014) | | | | |
| MsTMIP-v2 (SG3) | VEGAS | Zeng et al. (2005) | X | X | X | X |
| TRENDY-v9 | CLASSIC | Melton et al. (2020) | | | | |
| TRENDY-v9 | IBIS | Yuan et al. (2014) | | | | |
| TRENDY-v9 | ISBA CTRIP | Delire et al. (2020) | | | X | |
| TRENDY-v9 | JSBACH | Reick et al. (2021) | | | | |
| TRENDY-v9 | JULES ES 1P0 | Clark et al. (2011) | | | | |
| TRENDY-v9 | LPJ | Thonicke et al. (2001, 2010), Venefsky et al. (2002) | | | | |
| TRENDY-v9 | LPX-Bern | Lienert and Joos (2018) | X | | | |
| TRENDY-v9 | OCN | Zaehle and Friend (2010) | | | X | |
| TRENDY-v9 | ORCHIDEE | Yue et al. (2014) | X | X | X | X |
| TRENDY-v9 | ORCHIDEE CNP | Goll et al. (2017) | | | | |
| TRENDY-v9 | ORCHIDEEv3 | Vuichard et al. (2019) | | | X | |
| TRENDY-v9 | SDGVM | Walker et al. (2017) | | | X | |
| TRENDY-v9* | CLM | Lawrence et al. (2019) | X | | | |
| TRENDY-v9* | ISAM | Meiyappan et al. (2015) | X | | | |
| TRENDY-v9* | VISIT | Ito (2010); Kato et al. (2013) | | | | |
| Inversions | CAMS v20r1 | Bergamaschi et al. (2007, 2009), Chevallier et al. (2010) | X | X | X | X |
| Inversions | CT2019B | Jacobson et al. (2020) | X | X | X | X |
| Inversions | CTE 2020 | Peters et al. (2007, 2010), van der Laan-Luijkx et al. (2017) | X | X | X | X |
| Inversions | CarbonTracker-Lagrange | Hu et al. (2019) | X | X | X | X |
| Inversions | GIM | Shiga et al. (2018a) | X | X | X | X |
| Inversions | Jena sEXTocNEET | Rödenbeck (2005), Rödenbeck et al. (2018) | X | X | X | X |
| Inversions | MIROC | Patra (2018), Chandra et al. (2022) | X | | | |
| Inversions | NISMON | Niwa et al. (2017a, b), Niwa (2020) | X | | | |

b) and the Trends in Net Land-Atmosphere Exchange version 9 ensemble (TRENDY-v9; Friedlingstein et al., 2020; Sitch et al., 2015). MsTMIP is a model intercomparison project aimed at exploring the impact of structural differences in models by prescribing a fixed protocol with a semi-factorial design and consistent environmental driver data for an ensemble of models (Huntzinger et al., 2013, 2018, 2020; Wei et al., 2014a, b). We use the SG3 and BG1 simulations from MsTMIP-v2, where atmospheric $CO_2$ and land-use history are time-varying and nitrogen deposition rates are held constant in SG3, and land-use history, atmospheric $CO_2$, and nitrogen deposition are time-varying in BG1 (Table 1). MsTMIP-v2 defines NEE to be NEE $= R_h + R_a + F_{disturbance} + F_{product} - $ GPP where $R_h$ is heterotrophic respiration, $R_a$ is autotrophic respiration, $F_{disturbance}$ is the sum of fire and land use change fluxes, and $F_{product}$ is the decay of harvested wood products. Some models do not include fire, land use, and/or product fluxes, depending on the processes that are represented by the models. TRENDY includes simulations from an ensemble of dynamic global vegetation models (DGVMs) that are run annually as part of the Global Carbon Project yearly evaluation. TRENDY also includes a set of factorial simulations for the historical period (Friedlingstein et al., 2022). Here we use the S3 simulation from TRENDY-v9 where $CO_2$, climate, and land use forcings are time-varying. TRENDY defines NEE as NEE $= R_h + R_a + F_{fire} + F_{harvest} + F_{grazing} + F_{LUC} - $ GPP, where $F_{fire}$ are emissions from natural and human-sparked fires, $F_{harvest}$ is emissions from crop harvest, $F_{grazing}$ are fluxes from livestock grazing, and $F_{LUC}$ are fluxes resulting from land use change. Models within TRENDY-v9 vary in terms of which components are included. We detail the impact of varying NEE definitions in bottom-up models on our results in Sect. 3.1. Top-down estimates come from a set of inverse model estimates assembled in support of the REgional Carbon Cycle Assessment and Processes 2 (RECCAP-2) analysis (Ciais et al., 2022). RECCAP-2 is a project aimed at quantifying carbon budgets on regional scales across the globe.

For assessing consistency and agreement at biomes scales we use the same biome map as Shiga et al. (2018) TS7 and Sun et al. (2021) that is based on an International Geosphere-Biosphere Programme (IGBP) land cover classification map. MsTMIP-v2 imposes a consistent biome map for all models (Wei et al., 2014 TS8), while models from TRENDY-v9 use various sets of plant functional types (PFTs), resulting in differences in land cover representations (Seiler et al., 2022). While the TRENDY-v9 models have differences in land cover representation, they do use common land use and land cover change (LULCC) forcing data (Seiler et al., 2022). Though using the same biome map across all models would allow for greater standardization amongst TBMs, doing so is difficult due to model-specific setups. Understanding the impact of the various maps used by models is also difficult as few models provide outputs at the resolution necessary

to do a comprehensive analysis, such as evaluating whether specific PFTs are present at in situ observation sites (Seiler et al., 2022). However, Sun et al. (2021) did compare the impact of model-specific biome classifications for four models that provided PFT information at finer resolutions and showed that model-specific biome classification was not a primary driver of inter-model variability.

All models were re-gridded to a $1° \times 1°$ spatial and monthly temporal resolution and then cropped to a uniform North American domain for the study period of 2007–2010. For TBMs, only land fluxes are included in model output files and thus ocean grid cells are removed prior to regridding. This period was chosen because this was the time frame for which all models overlapped temporally (until 2010) and during which high-resolution atmospheric transport footprints (see Sect. 2.3) were available (starting in 2007) to link model estimates to atmospheric $CO_2$ observations. We use the model ensemble average when assessing agreement between bottom-up and top-down model ensembles. While this is a simple approach, Schwalm et al. (2015) found that the added complexity of skill-based integration does not materially change flux estimates based on TBM ensembles. The model simulations from MsTMIP-v2 and TRENDY-v9 ensembles were merged to create one TBM superensemble. Because MsTMIP-v2 and TRENDY-v9 have three models in common (CLM, ISAM, and VISIT; Table 1), the simulations from these models included in TRENDY-v9 were used in this analysis because TRENDY-v9 has more recent updates to models than MsTMIP-v2 (e.g., CLM5 vs. CLM4).

### 2.1.2 Atmospheric $CO_2$ observations

Atmospheric $CO_2$ observations are from ObsPack $CO_2$ GLOBALVIEWplus v3.2 (Cooperative Global Atmospheric Data Integration Project, 2017; Masarie et al., 2014) wherein continuous in situ observations were averaged to three-hourly averaged $CO_2$ measurements. Averaging is centered at 15:00 local time for most sites and 16:00 or 17:00 for a few sites; these times were chosen because afternoon observations are expected to have lower transport model errors stemming from model representation of planetary boundary layer dynamics (Lin et al., 2017). Urban sites were excluded as this analysis focuses on the biospheric signal. From the 44 continuous-monitoring towers for the 2007–2010 period selected here, there are around 57 700 available mid-afternoon observations, among which around 39 300 observations were used in the analysis and around 18 400 (32 %) observations were filtered out (Table S1). Data with extreme outliers and $CO_2$ enhancements above 30 ppmv are filtered out as described in Fang and Michalak (2015). In addition, data that have a large sensitivity to ocean fluxes (Gourdji et al., 2012) and data with potential transport model errors are also filtered out (Gourdji et al., 2012; Shiga et al., 2014 TS9).

To isolate the biospheric enhancement or drawdown, background $CO_2$ values and signals from fossil fuel emissions

were pre-subtracted. The impact of fossil fuel emissions on available $CO_2$ observations was calculated using footprints from a Lagrangian atmospheric transport model (see Sect. 2.3) and emissions from the Fossil Fuel Data Assimilation System (FFDAS v2; Asefi-Najafabady et al., 2014), scaled to $1° \times 1°$ spatial resolution and three-hourly temporal resolution to be consistent with the setup of atmospheric transport. The background $CO_2$ values (or boundary conditions) were calculated similarly to Jeong et al. (2013), where vertical profiles from aircraft data and marine boundary layer data are used to run back trajectories and the endpoints of the back trajectories are sampled to obtain background $CO_2$ values. Overall, data processing and filtering were done in a similar manner to Shiga et al. (2018a) and Sun et al. (2021).

### 2.1.3 Absorbed photosynthetically active radiation

We use absorbed photosynthetically active radiation (APAR) as a baseline for assessing model performance. APAR is a first-order driver of gross primary productivity (GPP) (Monteith, 1972). We chose to use APAR as opposed to a GPP product, such as MODIS GPP, because we wanted to use the simplest data-driven baseline possible. We view APAR as a simple and more direct baseline than MODIS GPP because MODIS GPP itself is modeled using multiple parameters and is therefore itself a type of model (Running and Zhao, 2015). Because NEE is the balance between GPP and ecosystem respiration, and ecosystem respiration is highly correlated with GPP (Janssens et al., 2001; Baldocchi, 2008), we expect APAR to explain a portion of the variability in NEE. Given that remotely sensed APAR in and of itself does not incorporate biochemical processes governing gas exchange, which are commonly represented in TBMs, we would therefore expect models to outperform APAR in explaining observed variability in atmospheric $CO_2$ concentrations. APAR is calculated as the product of MODIS fAPAR (Myneni et al., 2002, 2015) and photosynthetic active radiation (PAR) following Sun et al. (2021). PAR is calculated by rescaling shortwave radiation from the North American Regional Reanalysis dataset (Mesinger et al., 2006) following the empirical relationship from Meek et al. (1984).

### 2.1.4 Flux tower NEE

We qualitatively compared FLUXNET2015 NEE with $1° \times 1°$ modeled NEE estimates. FLUXNET2015 is a global data product for eddy covariance measurements of carbon, water, and energy exchange between the atmosphere and biosphere (Pastorello et al., 2021). We used the NEE data product with the variable USTAR threshold (VUT), where USTAR (i.e., friction velocity) thresholds vary yearly. We used data from five flux tower locations that are within the same $1° \times 1°$ grid cell as towers from ObsPack $CO_2$ GLOBALVIEWplus v3.2 that are used to evaluate seasonality (see Sect. 2.3). The three flux towers located in the same $1° \times 1°$ grid cell as the AME

tower (Mead, Nebraska; Miles et al., 2012) are located at the University of Nebraska Agricultural Research and Development Center near Mead, Nebraska. The three sites are an irrigated continuous maize site (US-Ne1; Suyker, 2016a), an irrigated maize–soybean rotation site (US-Ne2; Suyker, 2016b), and a rainfed maize–soybean rotation site (US-Ne3; Suyker, 2016c). The two flux towers located in the same $1° \times 1°$ grid cell as the LEF tower (Park Falls, Wisconsin; Andrews et al., 2014) are Park Falls (US-Pfa; Desai, 2016a) and Willow Creek (US-WCr; Desai, 2016b), Wisconsin. Given the scale mismatch between the footprint of flux tower observations ($\sim 1 \text{ km}^2$) and the resolution of the models examined here ($1° \times 1°$), comparisons are only interpreted qualitatively.

### 2.2 Determining consistency across spatial scales

Bottom-up and top-down NEE estimates are compared to determine which modeling approach provides the more consistent estimate at various spatial scales. Consistency is quantified as the standard deviation across model estimates in the ensemble of TBMs and across the ensemble of inversions. We assess consistency at nested scales of $1° \times 1°$, $3° \times 3°$, $5° \times 5°$, $7° \times 7°$, and $9° \times 9°$ for all grid cells throughout the North American domain by first calculating the area-weighted average NEE for each model within an ensemble and then calculating the ensemble standard deviation at each scale (Figs. 2, S1). We also assess consistency at the biome and continental scale. We then compare the consistency of TBMs to that of inversions at each scale to determine whether the more consistent approach is scale-dependent. We use an $F$ test to determine whether differences in consistency between the two ensembles is statistically significant ($p < 0.05$).

### 2.3 Evaluation against atmospheric observations

We assess the degree to which model-simulated NEE estimates can reproduce basic aspects of observed atmospheric $CO_2$ concentrations using three sets of metrics focusing on variability, magnitude, and seasonality. We compare observed atmospheric $CO_2$ concentrations with modeled $CO_2$ concentrations for all models in the TBM and in the inversion ensembles.

The sensitivity of $CO_2$ enhancements at available observation locations and times to upwind fluxes (ppm $[\mu\text{mol m}^{-2}\text{ s}^{-1}]^{-1}$) are represented using the Stochastic Time-Inverted Lagrangian Transport (STILT) model (Lin et al., 2003; Nehrkorn et al., 2010) driven by meteorological fields simulated by the Weather Research and Forecasting (WRF) model (Skamarock and Klemp, 2008) for North America and aggregated to a $1° \times 1°$ spatial resolution and three-hourly temporal resolution. Footprints were generated as part of the NOAA CarbonTracker-Lagrange regional inverse modeling framework (Hu et al., 2019; https://gml.noaa.gov/ccgg/carbontracker-lagrange/, last access: TS10). The footprints used here cover the time period

of 2007–2010. The WRF-STILT model has been previously used to assess TBM estimates of $CO_2$ fluxes (Fang et al., 2014; Fang and Michalak, 2015; Sun et al., 2021) and to quantify greenhouse gas fluxes (Gourdji et al., 2012; Jeong et al., 2013; Miller et al., 2014; Shiga et al., 2018a). Here we use the footprints to translate the space–time patterns of carbon fluxes into their impacts on atmospheric $CO_2$ observations in order to assess the model estimates' ability to represent specific aspects of atmospheric $CO_2$ variability in space and time.

We use the coefficient of determination ($R^2$) between observed and modeled $CO_2$ drawdowns or enhancements as the metric for explained variability, i.e., quantifying the degree to which model estimates can reproduce observed spatiotemporal variability across all observation locations and times. We use the transported signal based on spatiotemporal variability in APAR as a lower benchmark for model performance (see Sect. 2.1.3). If a model has an $R^2$ value that is lower than APAR's $R^2$ value, then that model is removed from the ensemble when this metric is applied (Fig. S2).

Similarly, we use the root mean squared error (RMSE) between observed and modeled $CO_2$ drawdowns or enhancements as the metric for the magnitude of $CO_2$ signals from modeled fluxes. We again use APAR as a lower benchmark, but in this case, we first rescale APAR by minimizing RMSE. This step comes down to performing a linear regression between the transported APAR signal and the observed $CO_2$ enhancements, which also implicitly embodies the necessary unit conversion. If a model has a higher RMSE than that of the rescaled APAR signal, then that model is removed from the ensemble when this metric is applied (Fig. S3).

We use four sub-metrics to assess seasonality. Seasonality is assessed at individual towers that have $CO_2$ concentration observations for at least 50 % of days in the study period and for which the maximum data gap is less than 31 consecutive days. Only four towers within our study region meet these criteria (red symbols in Fig. 1): LEF (Park Falls, Wisconsin, USA), AME (Mead, Nebraska, USA), WKT (Moody, Texas, USA), ETL (East Trout Lake, Saskatchewan, Canada). The towers included in this analysis fall within different biomes and have different average seasonal cycles, allowing for assessment of agreement between modeled and observed $CO_2$ seasonal cycles across various landscapes. We also conducted a sensitivity analysis to test if the number of towers used for the seasonality metric has a significant impact on model selection. To do so, we relaxed the criteria for tower selection to a maximum allowable data gap of 80 consecutive days yielding four additional towers: SGP (Southern Great Plains, OK, USA), AMT (Argyle, ME, USA), OFR (Fir, OR, USA), and EGB (Egbert, ON Canada) (see Fig. S4). We calculate the monthly average observed and modeled $CO_2$ seasonal cycle for each of these four towers across the four years (2007–2010).

The first two seasonality sub-metrics are the $R^2$ and RMSE between the monthly averaged seasonal cycles of ob-

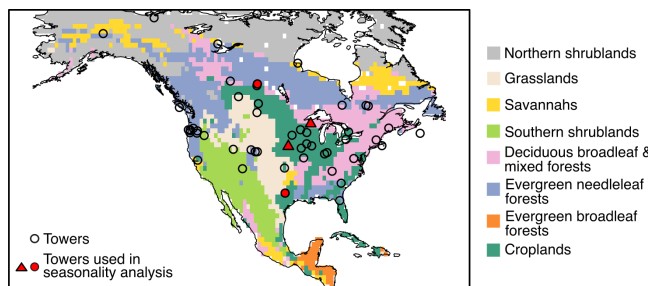

**Figure 1.** Map of biomes in North America with the locations of continuous-monitoring towers used in this study. Symbols represent the locations of towers in the $CO_2$ observational network. Red filled symbols represent the subset of towers with high temporal coverage that were used to evaluate how well model-simulated NEE estimates reproduce the seasonality of atmospheric $CO_2$ observations, whereas all towers are used for the magnitude and variability metrics (see Sect. 2.3). Red triangles represent locations of towers with high temporal coverage where there are also eddy covariance flux towers nearby (see Fig. 2).

served $CO_2$ drawdowns or enhancements and of $CO_2$ drawdowns or enhancements resulting from the transported carbon fluxes at each of the four tower locations, for each model. We again use transported signals resulting from spatiotemporal patterns of APAR as a lower benchmark for model performance. A model with an $R^2$ value greater than that of APAR and an RMSE value less than the RMSE value for rescaled APAR is considered to meet the sub-metrics of seasonal variability and magnitude, respectively.

The third sub-metric is the amplitude of the seasonal cycle, which is defined as the difference between the maximum and minimum monthly averaged $CO_2$ concentrations in the average seasonal cycle (Zhao et al., 2016). The model-estimated amplitude of the seasonal cycle is evaluated at each of the four tower locations. We use the amplitude of the seasonal cycle from rescaled APAR as a lower baseline; because the seasonal cycle of APAR is less peaked than that of NEE, the same will be true for the seasonal cycle of the transported signal based on APAR relative to observed $CO_2$ enhancements (Figs. S5, S6). If the amplitude estimated from a model is greater than the amplitude of the transported and rescaled APAR signal, then that model is considered to meet the minimum threshold for the amplitude sub-metric.

The fourth seasonality sub-metric is the timing of peak uptake for the monthly averaged $CO_2$ concentrations, defined as the month when peak uptake occurs. If the predicted peak uptake falls within one month of the observed peak uptake in atmospheric $CO_2$, then the model is considered to pass based on this sub-metric. When applying the overall seasonality metric, a model is kept in the ensemble if it can meet the lower benchmark for at least two of the seasonality sub-metrics at all four tower locations, but the sub-metrics that it meets can vary from tower to tower.

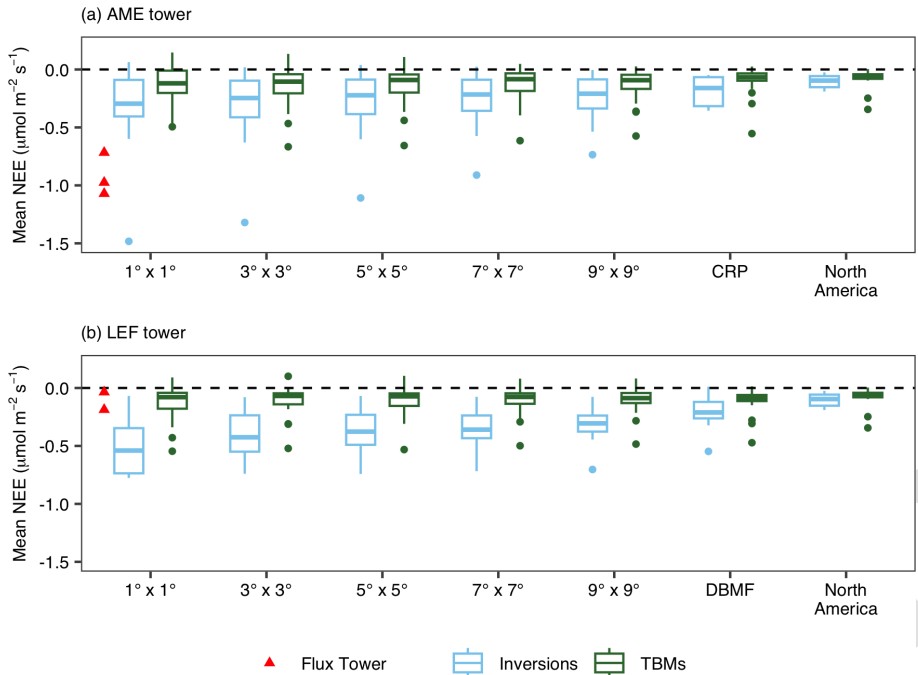

**Figure 2.** Example of consistency in atmospheric inversion and TBM ensembles across spatial scales centered at two grid cells located in the cropland (CRP) and deciduous broadleaf and mixed forest (DBMF) biomes. The AME tower, along with the US-Ne1, US-Ne2, and US-Ne3 flux tower sites, falls within the CRP biome. The LEF tower, along with the US-PFa and US-WCr flux towers, falls within the DBMF biome.

## 3 Results and discussion

### 3.1 Model consistency across scales

We find that the full ensemble of TBMs has more consistent carbon flux estimates across all examined spatial scales relative to inversions (Fig. 3a). This is evidenced by TBMs having a smaller standard deviation across the model ensemble for the majority of locations across North America, although there are a few regions where the ensemble of inversions has a smaller standard deviation or where the ensemble with the smaller standard deviation depends on the scale being examined. This result also holds at the biome and continental scale (Fig. 5 "all models"). This result is surprising, because atmospheric inversions are informed by large-scale atmospheric constraints, while TBMs are primarily constrained by process-based understanding derived at fine scales (e.g., plot scale). One would therefore expect that inversions would be more consistent at larger scales than TBMs.

However, the difference in the degree of consistency between inversions and TBMs is not statistically significant ($p > 0.05$) for large portions of the continent (Fig. 3b). This is because both ensembles have very high inter-model spread (see examples in Fig. 2), reducing the statistical significance of their differences. These results underscore the importance of statistical significance testing in interpreting model differences.

Earlier studies comparing bottom-up and top-down models have shown varying results in terms of bottom-up versus top-down model consistency for North America. Hayes et al. (2012) reported an average annual NEE estimate for North America of $-931 \pm 670\,\mathrm{TgC\,yr^{-1}}$ (mean $\pm 1$ standard deviation) across seven inverse models and $-511 \pm 729\,\mathrm{TgC\,yr^{-1}}$ across 12 TBMs for the period of 2000–2006, indicating a slightly higher consistency (lower standard deviation) across inversions at the continental scale. King et al. (2015), on the other hand, reported a mean $\pm 1$ standard deviation annual net land–atmosphere exchange for North America of $-890 \pm 409\,\mathrm{TgC\,yr^{-1}}$ across 11 inverse models and $-364 \pm 120\,\mathrm{TgC\,yr^{-1}}$ across 10 TBMs for the period 1990–2009, indicating that TBMs were substantially more consistent. The synthesis presented in the State of the Carbon Cycle 2 (SOCCR-2) report further confirmed that the relative consistency among top-down versus bottom-up models varies across studies (Hayes et al., 2018). These earlier studies not only used older versions of model simulations than examined here, but also included far fewer TBMs. At the global scale, the most recent assessment shows that TBMs have a greater model spread (i.e., lower consistency) than do inversions (Friedlingstein et al., 2022). Though these studies are primarily focused on assessing agreement between bottom-up and top-down methods at a single large spatial scale, they demonstrate the difficulty in assessing consistency when ensemble size is limited, and statistical significance is not evaluated. In addition, comparing consistency of bottom-up

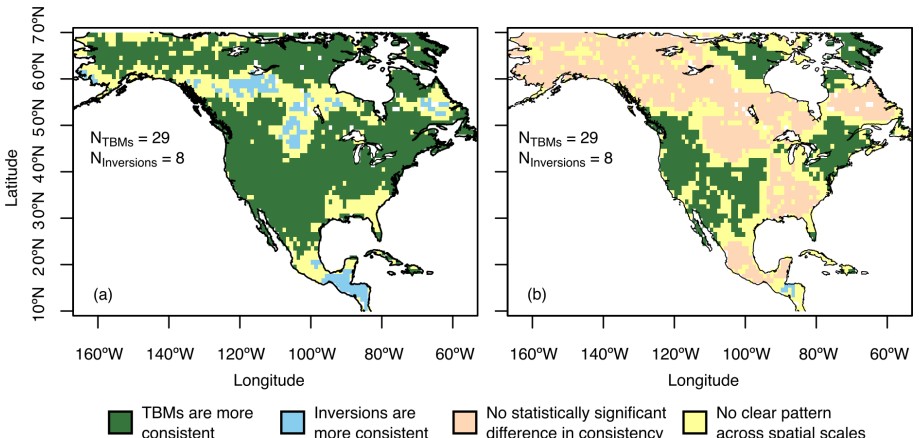

**Figure 3.** Maps showing whether the TBM or the atmospheric inversion ensemble has the more consistent NEE estimates across spatial scales. Maps show where each ensemble has the most consistent estimate (smallest standard deviation) at each of the following scales: $1° \times 1°$, $3° \times 3°$, $5° \times 5°$, $7° \times 7°$, and $9° \times 9°$. Panel **(a)** shows the most consistent ensemble when the statistical significance of the difference in consistency is not taken into account. Panel **(b)** shows the result when statistical significance is taken into account. Green regions represent where TBMs have the smaller standard deviation at every examined scale, while blue regions show where inversions are more consistent. Orange regions represent areas where there is no statistically significant difference in consistency at any spatial scale. Yellow regions represent areas where there are inconsistencies across spatial scales. More specifically, in panel **(a)**, yellow regions represent areas where TBMs are more consistent at some scale while inversions are more consistent at other scales, whereas in panel **(b)** yellow regions represent areas where there is either a mix of statistically significant and not statistically significant differences across spatial scales or where there is a statistically significant difference across all scales, but neither inversion nor TBMs are more consistent across all scales.

and top-down models without assessing the statistical significance of observed differences may lead to misleading conclusions. Kondo et al. (2020) found TBMs to have a smaller inter-model spread in regional budget estimates, but the large spread in the seasonality of carbon uptake for TBMs made it difficult to deduce whether bottom-up models are more reliable than top-down models based on consistency alone.

While the consistency of model ensembles can be used as one measure of uncertainty in modeling carbon uptake, both bottom-up and top-down methods carry uncertainties that may not be fully captured by ensemble spread alone. For example, the use of satellite data to augment data coverage for atmospheric inversions did not clearly improve consistency in inverse-model-based estimates in a recent intercomparison study (Crowell et al., 2019). Moreover, the differences in the processes incorporated in the various bottom-up models may impact the consistency across the TBM ensemble. We examined how using a simple definition of NEE ($R_h + R_a - $ GPP) impacted consistency within the MsTMIP-v2 ensemble and found that estimates from the MsTMIP-v2 ensemble are less consistent across models when using only GPP, $R_h$, and $R_a$ in the calculation of NEE (Fig. S7). In other words, the models are more consistent when they include other components of NEE, even though those components differ from model to model. This seems to suggest that models may implicitly target a presumed net land sink irrespective of the processes included. This appears to be in line with earlier research that found some TBMs arrive at similar estimates of carbon uptake even though they show large disagreements

on the primary driver of increased uptake in recent decades, while other TBMs arrive at dissimilar estimates despite having similar sensitivities (Huntzinger et al., 2017).

## 3.2 Impact of variability, magnitude, and seasonality on consistency

We evaluate whether greater consistency corresponds to greater accuracy by subsetting the model ensembles using metrics based on the variability, seasonality, and magnitude of atmospheric observations (see Sect. 2.3). Limiting the bottom-up ensemble to only include TBMs that reproduce the variability of atmospheric observations better than APAR reduced the ensemble size from 29 to 9 models. In other words, over two-thirds of TBMs represented the space–time variability of atmospheric $CO_2$ less well than did APAR (Table 1, Fig. S2). Conversely, all inversions performed better than this benchmark (Table 2), which is expected given that the inversions use all or a subset of the same observations in estimating fluxes. Sub-selecting models that could represent aspects of the seasonality of atmospheric observations reduced the ensemble of TBMs from 29 to 10 and the ensemble of inversions from 8 to 6. Limiting the ensembles to models that could represent the magnitude of observed atmospheric $CO_2$ signals reduced the ensemble of TBMs from 29 to 4, while the ensemble of inversions was reduced from 8 to 6. Only three TBMs (and six inversions) remained when all three metrics were applied. For the three TBMs that are in both the MsTMIP-v2 and TRENDY-v9 ensem-

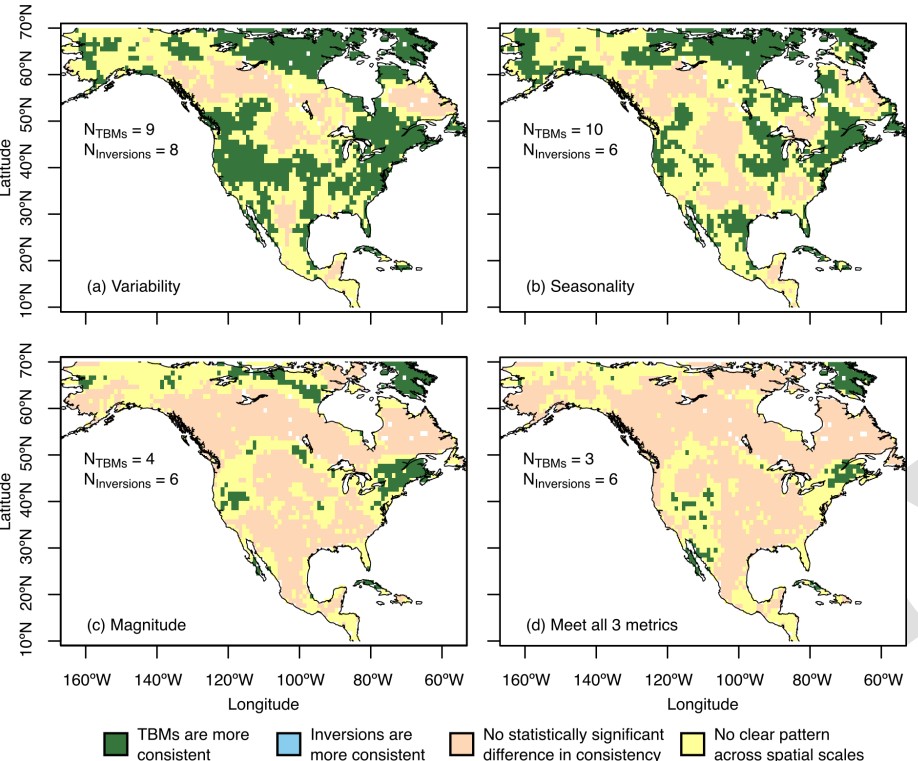

**Figure 4.** TS11 Maps showing where the TBM or the atmospheric inversion ensemble has the more consistent NEE estimates across spatial scales when the ensembles are limited to those models that meet **(a)** variability, **(b)** seasonality, **(c)** magnitude, or **(d)** all three metrics. Colors are as defined in Fig. 3.

bles, we retained the TRENDY-v9 simulations as described in Sect. 2.1.1. Had we included the three MsTMIP-v2 simulations, however, none of them met any of the three metrics and they therefore would have been excluded from the sub-
5 sets meeting the variability, seasonality, and magnitude metrics.

The large reduction in the ensemble size when even basic benchmarks related to observed atmospheric $CO_2$ signals are applied (i.e., filtering based on a minimum accuracy thresh-
10 old) indicates that ensemble spread (i.e., consistency) is unlikely to be a good indicator of actual uncertainty in our understanding of North American carbon balance. An example of consistency not necessarily capturing uncertainty is shown in Fig. 2 at the $1° \times 1°$ scale where TBMs and inversions
15 show better agreement with eddy covariance flux tower observations in the deciduous broadleaf and mixed forest biome (Fig. 2b) than in the cropland biome (Fig. 2a) despite having similar model consistency in both biomes. It is unclear whether the models and observations disagree in the crop-
20 land biome due to sub-grid scale heterogeneity (Melton and Arora, 2014) versus inaccuracies in the models (Schuh et al., 2014; Guanter et al., 2014; Sun et al., 2021), but this comparison nevertheless illustrates how consistency may not capture the full extent of uncertainty in model simulations.

Fang et al. (2014) also noted that atmospheric observations 25 can be used to evaluate flux patterns from TBMs in terms of models' ability to explain atmospheric observations. Using a similar approach here we see that differentiating between models that reproduce basic features of atmospheric $CO_2$ signals leads to a shift from TBMs having the smaller ensemble 30 standard deviation across scales to there being no statistically significant difference between the ensemble standard deviations of TBMs and inversions across most of the continent. This is particularly true when models are selected based on consistency with all three metrics. 35

Applying the variability, magnitude, and seasonality metrics reduces the areas in North America for which TBMs have a statistically significantly greater consistency than do inversions (Fig. 4). Once all three metrics are applied, almost no areas remain where TBMs have a higher consis- 40 tency across all scales. This result is enlightening, because it indicates that once basic aspects of *atmospheric* observational constraints are taken into account, apparent differences in consistency between approaches disappear. The large reduction in the TBM ensemble size is part of the reason for 45 this change, so this result must be interpreted with caution. In other words, this result is less a product of the consistency of remaining inversions increasing and more a product of the statistical significance of differences in consistency being re-

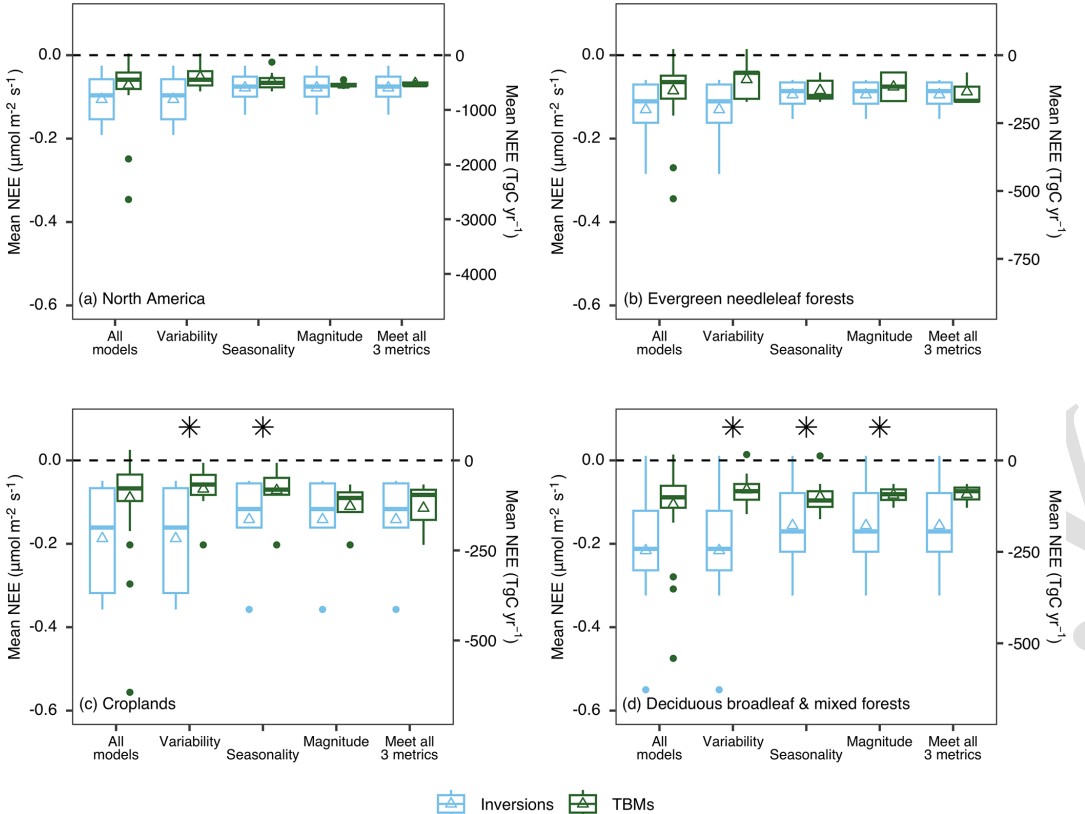

**Figure 5.** Estimated NEE for TBMs and atmospheric inversions for all models within their respective ensembles as well as subsets of the ensembles that meet the variability, seasonality, or magnitude metrics, or that meet all three. Boxplots represent the model-specific average NEE estimates across the models included in each ensemble; triangles represent the across-model mean. Panel **(a)** shows NEE for North America, while panels **(b)**–**(d)** show NEE for specific biomes. Stars represent cases for which there is a statistically significant difference ($p < 0.05$) in the consistency of the TBM versus the inversion ensemble.

duced due to the reduction in the sample size. This filtering exercise also indicates that, although the ensemble of inversions examined here has lower consistency, it may actually exhibit higher accuracy as evidenced by the smaller reduction in ensemble size and higher number of models that meet all three criteria.

### 3.3 Implications for understanding of North American carbon balance

#### 3.3.1 Impact on model consistency

The impact of applying accuracy metrics to the ensembles offers a glimpse into the true North American carbon sink. While this approach did increase model consistency for TBMs, it did not impact model consistency for atmospheric inversions. The lack of increase in the consistency of inversions once accuracy metrics are applied indicates that there is a wide range of overall flux patterns and magnitudes that are consistent with large-scale atmospheric constraints. It is interesting, therefore, that model consistency among TBMs does increase substantially when accuracy metrics are applied (Fig. 5), although again this has to be interpreted with caution given the small number of remaining models. This contrast implies that, once accuracy metrics are applied, remaining TBMs all reproduce observed features of atmospheric observations using similar flux patterns, while remaining inversions reproduce observed features of atmospheric observations using a wider range of fluxes. The large spread of inversions that are also consistent with the same atmospheric constraints indicates that consistency across model ensembles is likely a poor indicator of overall uncertainty. This suggests that decreases in model spread do not necessarily indicate increases in model accuracy, as has also been suggested in previous studies (Annan and Hargreaves, 2010; Knutti et al., 2010; Kondo et al., 2020; Lovenduski and Bonan, 2017). Because TBMs were already informed by process-based understanding, subsetting the ensemble to those TBMs that are also consistent with broad features of atmospheric $CO_2$ observations may lead to the "best of both worlds". The high consistency of the remaining (albeit small number of) TBMs is a sign that these models are more likely to capture the true North American carbon sink.

**Table 2.** Impact of filtering ensembles based on key metrics derived from observational constraints. The mean $\pm$ standard deviation (median) fluxes (in units of $TgC\,yr^{-1}$) for inversions and TBMs when metrics are applied to North America and biomes with the largest data constraint across 2007–2010.

| Ensemble | Case | North America | Evergreen needleleaf forests | Croplands | Deciduous broadleaf and mixed forests |
|---|---|---|---|---|---|
| Inversions | All models | $-808 \pm 476\ (-734)$ | $-200 \pm 119\ (-170)$ | $-217 \pm 151\ (-186)$ | $-247 \pm 196\ (-242)$ |
| | Variability | $-808 \pm 476\ (-734)$ | $-200 \pm 119\ (-170)$ | $-217 \pm 151\ (-186)$ | $-247 \pm 196\ (-242)$ |
| | Seasonality | $-597 \pm 322\ (-574)$ | $-146 \pm 57\ (-132)$ | $-164 \pm 136\ (-135)$ | $-178 \pm 137\ (-194)$ |
| | Magnitude | $-597 \pm 322\ (-574)$ | $-146 \pm 57\ (-132)$ | $-164 \pm 136\ (-135)$ | $-178 \pm 137\ (-194)$ |
| | Meet all three | $-597 \pm 322\ (-574)$ | $-146 \pm 57\ (-132)$ | $-164 \pm 136\ (-135)$ | $-178 \pm 137\ (-194)$ |
| TBMs | All models | $-547 \pm 519\ (-449)$ | $-131 \pm 109\ (-99)$ | $-104 \pm 128\ (-78)$ | $-121 \pm 113\ (-101)$ |
| | Variability | $-401 \pm 225\ (-449)$ | $-90 \pm 67\ (-65)$ | $-79 \pm 68\ (-67)$ | $-80 \pm 50\ (-84)$ |
| | Seasonality | $-475 \pm 161\ (-505)$ | $-130 \pm 41\ (-150)$ | $-83 \pm 61\ (-81)$ | $-100 \pm 49\ (-110)$ |
| | Magnitude | $-542 \pm 70\ (-551)$ | $-117 \pm 61\ (-116)$ | $-128 \pm 74\ (-104)$ | $-95 \pm 28\ (-93)$ |
| | Meet all three | $-517 \pm 59\ (-549)$ | $-134 \pm 61\ (-168)$ | $-133 \pm 90\ (-86)$ | $-93 \pm 33\ (-84)$ |

### 3.3.2 Impact on model agreement

Most promisingly, not only does the difference in consistency between top-down and bottom-up models decrease when ensembles are filtered based on key metrics derived from observational constraints, but the agreement between top-down and bottom-up models improves dramatically as well (Fig. 5a). Indeed, once all three metrics are applied, both the mean and the median across the remaining TBMs are very close to the mean and the median across the remaining inversions. In other words, filtering leads to better agreement between top-down and bottom-up estimates of North American carbon balance, a goal that has proven elusive up to now. The mean (median) North American fluxes across 2007–2010, when all three criteria are applied, is $-517\,TgC\,yr^{-1}$ ($-549\,TgC\,yr^{-1}$) for the TBMs and $-597\,TgC\,yr^{-1}$ ($-574\,TgC\,yr^{-1}$) for the inversions (Table 2).

Agreement for the three biomes that are best-constrained by atmospheric observations is also improved (Fig. 5b, c, d) although to a lesser extent. These biomes are evergreen needleleaf forests, croplands, and deciduous broadleaf and mixed forests biomes, based on an analysis by Sun et al. TS12 that showed these biomes to have the greatest sensitivity of atmospheric observations to fluxes (2023). When the seasonality metric is recalculated using eight towers selected from the sensitivity analysis (see Sect. 2.3), there are minimal changes to agreement between bottom-up and top-down estimates. The deciduous broadleaf and mixed forests, which is the biome with the lowest observational constraint of the three biomes, showed decreased agreement suggesting that using towers with high temporal data availability, rather than more towers, is important for capturing seasonality. The mean (median) uptake for evergreen needleleaf forests when all three metrics are applied is $-134\,TgC\,yr^{-1}$ ($-168\,TgC\,yr^{-1}$)

for the TBMs and $-146\,TgC\,yr^{-1}$ ($-132\,TgC\,yr^{-1}$) for the inversions (Table 2). For croplands, the mean (median) is $-133\,TgC\,yr^{-1}$ ($-96\,TgC\,yr^{-1}$) for the TBMs and $-164\,TgC\,yr^{-1}$ ($-135\,TgC\,yr^{-1}$) for the inversions (Table 2). The mean (median) for deciduous broadleaf and mixed forests is $-93\,TgC\,yr^{-1}$ ($-84\,TgC\,yr^{-1}$) for the TBMs and $-178\,TgC\,yr^{-1}$ ($-194\,TgC\,yr^{-1}$) for the inversions (Table 2).

Lower improvement in the agreement between TBMs and inversions at the biome scale relative to the continental scale may have resulted from disagreement on where major sinks in North America lie. However, to understand why models disagree, it is necessary to understand what gives rise to differences. For example, deciduous broadleaf and mixed forests were found to account for the majority of the interannual variability in NEE for North America when using a top-down approach, but TBMs disagreed on whether forested or non-forested biomes contribute most strongly to interannual variability and what the primary environmental drivers of this variability are (Shiga et al., 2018b). TBMs have also been shown to have greater interannual variability in western temperate North America than in eastern temperate North America, albeit with substantial model spread (Byrne et al., 2020). A recent study found that TBMs that did well at reproducing observed $CO_2$ variability exhibited substantially stronger growing-season carbon uptake in croplands relative to other models (Sun et al., 2021). These studies highlight that there is still uncertainty about the geographic distribution, interannual variability, and climatic drivers of North American carbon uptake. This likely plays a role in the reduced agreement at the biome scale relative to the continental scale observed here.

### 3.3.3 Sensitivity analyses and study limitations

One potential reason for the lack of convergence between bottom-up and top-down approaches is the limited number of towers used for the seasonality metrics. These towers are primarily located in the midcontinent (Fig. 1), although observations from towers are also influenced by fluxes in other regions (see, e.g., Sun et al., 2023, Extended Data Fig. 2 CEI). To test this this hypothesis, we conducted a sensitivity analysis to explore the impact of including four additional towers (see Sect. 2.3). We found that including more towers in other parts of North America did not change our primary findings. Specifically, we found that 10 models perform well based on the seasonality metric irrespective of which subset of towers is used (i.e., original four towers, additional four towers, or all eight towers), while six and four additional models also meet the metric when the original and additional sets of four towers are used, respectively. Though modifying the towers used impacts the consistency of ensembles based on the seasonality metric alone, the impact on the consistency and agreement of models that meet all three metrics is minimal (Fig. S8).

While increased data availability in biomes that are not well sampled would allow for a more robust evaluation of models' ability to capture key aspects of seasonality, several studies have found that even in areas with high data availability there is still noticeable disagreement between inversions (Bastos et al., 2020; Kondo et al., 2020). This indicates that model-specific issues may be more important than data availability in explaining discrepancies between bottom-up and top-down estimates. In the context of inverse modeling, the choice of fossil fuel inventory and transport model have been identified as significant sources of uncertainties, though there is not a clear consensus as to what the primary source of uncertainty is (Bastos et al., 2020; Gaubert et al., 2019; Peylin et al., 2011; Schuh et al., 2019). For TBMs, noteworthy uncertainties stem from how (and whether) models represent key processes such as land use change, wood and crop harvesting, fire, and vegetation dynamics (Ahlström et al., 2015; Bastos et al., 2020; Hardouin et al., 2022; Tharammal et al., 2019).

Several studies also highlight lateral fluxes as a significant source of $CO_2$ emissions, yet many TBMs do not account for them (Drake et al., 2018; Kondo et al., 2020; Raymond et al., 2013). To test the possible impact of lateral fluxes on our analysis, we examined the effect of including lateral fluxes in bottom-up estimates on the agreement between bottom-up and top-down methods. To do so, we added gridded lateral fluxes of river export, crop trade, and wood trade to TBMs to make them more comparable with fluxes seen by inversions. We used two different estimates of river export from Byrne et al. (2023). The first is the gridded product, which incorporates results from the Global NEWS model (Byrne et al., 2022). The second is the same gridded product rescaled, so total river export equals the country total

river exports reported in Byrne et al. (2023), which is the mean of two models estimates (Global NEWS and DLEM). The differences in these two river export estimates highlight some of the uncertainties associated with estimating lateral fluxes (Byrne et al., 2023; Drake et al., 2018). We find that at the North American scale, incorporating lateral fluxes improved agreement between bottom-up and top-down models somewhat, but the change was not sufficient to explain discrepancies for the best-constrained biomes (Fig. S9). In the deciduous broadleaf and mixed forests and cropland biomes, lateral fluxes only partially explain discrepancies and applying the seasonality, variability, and magnitude metrics still improved agreement between bottom-up and top-down estimates (Fig. S9c–d). This contrasts with the evergreen and needle leaf forest biome where the inclusion of lateral fluxes led to better agreement between inversions and TBMs for the subset with all models included but ultimately exacerbated differences once models that meet all three criteria were selected (Fig. S9b). This aligns with the findings of Kondo et al. (2020) that accounting for lateral fluxes and using a standardized net $CO_2$ flux definition was not sufficient to explain discrepancies between inversions and TBMs at regional scales.

Next we explored whether agreement between top-down and bottom-up estimates was improved when a more consistent definition of NEE was applied to TBMs. To do so, we examined how using a simple definition of NEE ($R_h + R_a -$ GPP) impacted agreement between MsTMIP-v2 models and inversions; we found that agreement with inversions improved slightly (Fig. S7). Of the MsTMIP-v2 models we looked at, only four included both disturbance and product fluxes in their definition of NEE (CLM, CLM4VIC, TEM6, VEGAS), and how well these models agree with inversions varies by biome (Fig. S7). Bastos et al. (2020) found that models, irrespective of their inclusion of fire dynamics, exhibited similar performance, which could be attributed to heightened sensitivity of decomposition to temperature in models without fire. Moreover, Huntzinger et al. (2017) found TBMs can yield similar estimates despite diverging on their primary drivers. We therefore found that it is unlikely that varying NEE definitions provide a comprehensive explanation for the observed disparities.

There are additional sources of uncertainty that the sensitivity analyses described here cannot address. For example, the non-uniform spatial distribution of observation data throughout the domain limits the biomes that can be examined. Additional observations, particularly in regions that are not well sampled, would allow for additional analyses and would further increase the robustness of the continental-scale results. Leveraging larger ensembles of TBMs and inversions and incorporating an assessment of the impact of transport model choice would also allow for additional insight into the robustness of our findings. Though we showed that TBM estimates of North American carbon sink were actually more consistent when varying NEE definitions were used and that

discrepancies between bottom-up and top-down models cannot be fully resolved by the incorporation of lateral fluxes, improved standardization across TBMs in terms of represented processes would allow for a more accurate comparison with inversions. Finally, it is important to recognize that better performance under present climate conditions does not necessarily translate to better model performance under future conditions.

## 4   Conclusions

Comparing estimates from bottom-up and top-down methods across spatial scales and evaluating estimates in light of atmospheric $CO_2$ observations is useful for exploring persistent differences between these approaches. We show that the difference in consistency between bottom-up and top-down ensembles is not statistically significant for large regions of North America because of large variability within both ensembles, highlighting the importance of significance testing in interpreting model differences. We also find that ensemble spread is unlikely to be a good indicator of overall uncertainty in the North American carbon balance. This is because when the same benchmarks based on observed atmospheric $CO_2$ signals are applied to both ensembles, inversions use a wider range of fluxes than TBMs to reproduce observed atmospheric observation features. Encouragingly, once models are sub-selected based on their ability to reproduce basic aspects of observed atmospheric $CO_2$ variability, seasonality, and magnitude, bottom-up and top-down estimates of North American carbon balance agree at the continental scale and for large biomes therein. Notably, these findings remained robust after several sensitivity analyses were performed. The convergence in flux estimates between top-down and bottom-up approaches demonstrates the usefulness of filtering models based on their agreement with even basic features of large-scale observational constraints for assessing our understanding of carbon budgets. This finding is encouraging because it presents a promising path towards both improving model consistency and reducing uncertainties. Thus, continued efforts to reduce uncertainties should focus on improving consistency at scales finer than large continental domains and leveraging top-down observational constraints to refine understanding of the North American carbon balance.

*Data availability.* All data used are publicly available and the sources are provided in Sect. 2 and Tables 1 and S1.

*Supplement.* The supplement related to this article is available online at: https://doi.org/10.5194/bg-21-1-2024-supplement. TS13

*Author contributions.* AMM and KTF designed the study. WS, YPS, and KTF collected the data and prepared them for analysis. KTF prepared figures and wrote the manuscript with contributions from all authors.

*Competing interests.* The contact author has declared that none of the authors has any competing interests.

*Acknowledgements.* Wu Sun and Kelsey T. Foster acknowledges funding support by NASA through the Carbon Monitoring System (grant no. 80NSSC18K0165) and the Terrestrial Ecology programs (grant no. 80NSSC22K1253), with additional support from the Carnegie Institution for Science's endowment fund. Jiafu Mao was supported by the Terrestrial Ecosystem Science Scientific Focus Area (TES SFA) project funded by the US Department of Energy, Office of Science, Office of Biological and Environmental Research. Oak Ridge National Laboratory is supported by the Office of Science of the US Department of Energy under contract no. DE-AC05-00OR22725.

The authors thank Trevor Keenan and Xiangzhong Luo for processing and providing the MODIS FPAR data. We acknowledge the NCEP North American Regional Reanalysis data provided by the NOAA Physical Sciences Laboratory, Boulder, Colorado, USA (obtained from https://psl.noaa.gov/, last access: TS14). We thank all modelers and investigators who contributed to the Multi-scale synthesis and Terrestrial Model Intercomparison Project (MsTMIP; http://nacp.ornl.gov/MsTMIP.shtml, last access: TS15). Funding for the Multi-scale synthesis and Terrestrial Model Intercomparison Project activity was provided through NASA ROSES grant no. NNX10AG01A. Data management support for preparing, documenting, and distributing model driver and output data was performed by the Modeling and Synthesis Thematic Data Center at Oak Ridge National Laboratory (ORNL; https://nacp.ornl.gov, last access: TS16), with funding through NASA ROSES grant no. NNH10AN681. Finalized MsTMIP data products are archived at the ORNL DAAC (https://daac.ornl.gov, last access: TS17). The authors thank Stephen Sitch, Pierre Friedlingstein, and all modelers of the Trends in Net Land-Atmosphere Exchange project (TRENDY; https://blogs.exeter.ac.uk/trendy/, last access: TS18). We thank the following individuals for collecting and providing the atmospheric $CO_2$ data from the following sites: Arlyn Andrews for AMT, BAO, LEF, WBI, and WKT; Arlyn Andrews and Marc L. Fischer for WGC; Arlyn Andrews and Matt J. Parker for SCT; Arlyn Andrews and Stephan De Wekker for SNP; Sebastien Biraud and Margaret Torn for SGP; Tim Griffis for KCMP; Beverly Law, Andres Schmidt, and the TERRA-PNW group for data from the five Oregon sites OFR, OMP, OMT, ONG, and OYQ; Natasha Miles, Scott Richardson, and Ken Davis for AAC, ACR, ACV, AME, AOZ, FPK, RCE, RGV, RKW, RMM, and RRL; Britton Stephens and the Regional Atmospheric Continuous $CO_2$ Network in the Rocky Mountains (RACCOON) for HDP, NWR, RBA, and SPL; Colm

Sweeney for MVY; Kirk Thoning and Pieter Tans for BRW; Steven Wofsy and William Munger for HFM; Doug Worthy for BCK, BRA, CDL, CHM, EGB, ESP, EST, ETL, FSD, LLB, and WSA. Measurements at WGC were partially supported by grants from the California Energy Commission (CEC) Public Interest Environmental Research Program to the Lawrence Berkeley National Laboratory, which operates under US Department of Energy under contract no. DE-AC02-05CH11231. The authors thank the Atmospheric and Environmental Research, Inc. (AER) – particularly, Thomas Nehrkorn, John Henderson, and Janusz Eluszkiewicz – for conducting WRF-STILT simulations and providing transport footprints. We thank the CarbonTracker-Lagrange project team for proving the WRF-STILT transport footprints. The authors thank Kevin Gurney for FFDAS v2 data. We thank the CarbonTracker team for the CarbonTracker CT2019B results provided by the NOAA Global Monitoring Laboratory, Boulder, Colorado, USA (http://carbontracker. noaa.gov, last access: TS19). We thank the CarbonTracker-Lagrange team for terrestrial $CO_2$ fluxes data. We thank the CarbonTracker Europe team for the CarbonTracker Europe results provided by Wageningen University in collaboration with the ObsPack partners (http://www.carbontracker.eu, last access: TS20). We thank the Copernicus Atmosphere Monitoring Service (CAMS) team for the CAMS inversion results generated using Copernicus Atmosphere Monitoring Service Information (2020). Neither the European Commission nor ECMWF is responsible for any use that may be made of the information it contains. We thank Christian Rödenbeck for CarboScope-sEXTocNEET data (retrieved from http:// www.bgc-jena.mpg.de/CarboScope/?ID=sEXTocNEET_v4.3, last access: TS21). We thank the MIROC-ACTM team for the MIROC-ACTM inversions results that are provided by JAMSTEC (ArCS-II grant no. JPMXD1420318865, and ERTDF SII-8 grant no. JP-MEERF21S20800).

The computations presented here were conducted through Carnegie's partnership in the Resnick High Performance Computing Center, a facility supported by Resnick Sustainability Institute at the California Institute of Technology.

*Financial support.* TS22 This research has been supported by the National Aeronautics and Space Administration (grant nos. 80NSSC18K0165 and 80NSSC22K1253) and the Oak Ridge National Laboratory (grant no. DE-AC05-00OR22725).

*Review statement.* This paper was edited by Paul Stoy and reviewed by Guillermo Murray-Tortarolo and Xinyuan Wei.

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

**Remarks from the typesetter**

TS1    Please provide state abbreviation.

TS2    Please provide state abbreviation.

TS3    Please provide state abbreviation.

TS4    Please provide reference list entry.

TS5    Please provide reference list entry.

TS6    Please make sure the section references are correct here.

TS7    Do you mean "a" or "b"?

TS8    Do you mean "a" or "b"?

TS9    Please provide reference list entry.

TS10    Please provide date of last access (day month year).

TS11    Please note that Fig. 4 is not mentioned in the text.

TS12    Please provide year.

TS13    Please send a new Supplement as a \*.pdf without the title, authors, correspondence author, etc. as we will generate a Supplement title page during publication (with a citation including the DOI), which will contain this information.

TS14    Please provide date of last access (day month year).

TS15    Please provide date of last access (day month year).

TS16    Please provide date of last access (day month year).

TS17    Please provide date of last access (day month year).

TS18    Please provide date of last access (day month year).

TS19    Please provide date of last access (day month year).

TS20    Please provide date of last access (day month year).

TS21    Please provide date of last access (day month year).

TS22    Please note that there is a discrepancy between funding information provided by you in the acknowledgements and the funding information you indicated during manuscript registration, which we used to create this section. Please double-check your acknowledgements to see whether repeated information can be removed from the acknowledgements or changed accordingly. If further funders should be added to this section, please provide the funder names and the grant numbers. Thanks.

TS23    Please ensure that any data sets and software codes used in this work are properly cited in the text and included in this reference list. Thereby, please keep our reference style in mind, including creators, titles, publisher/repository, persistent identifier, and publication year. Regarding the publisher/repository, please add "[data set]" or "[code]" to the entry (e.g. Zenodo [code]).

TS24    Please provide pages or article number.

TS25    Please provide all authors.

TS26    Please provide DOI or ISBN.

TS27    Please provide date of last access (day month year).

TS28    Reference is missing in text.

TS29    Please provide all authors.

TS30    Please provide publisher and DOI or ISBN.

TS31    Reference is missing in text.

TS32    Please provide pages or article number with DOI.

TS33    Please provide publisher and DOI or ISBN.

TS34    Please provide date of last access (day month year).

TS35    Reference is missing in text.

TS36    Please provide all authors.

TS37    Please provide pages or article number.

TS38    Please provide journal name, volume and pages or article number with DOI.

TS39    Please provide DOI or ISBN.

TS40    Please provide DOI or ISBN.

TS41    Please provide DOI or ISBN.