# Peer review of "Multiscale assessment of North American terrestrial carbon balance"

_Biogeosciences, 2023_

## Author Response (AR1)

Point by point response to reviewers' comments

Reviewer # 1 (Remarks to the Author):
We thank the reviewer for their comments and suggestions. Please find our response below in **bold**.

[RC1-1] I had the pleasure of reviewing your manuscript, which I found very interesting and well written. Here you employed different measurements of variability to filter NEE from different models, showing that employing basic atmospheric constraints greatly improved consistency and reduces uncertainty. The approach is certainly promising and interesting, but also comes with several limitations that are not necessarily presented as they should. I believe the manuscript will be a strong contribution to the benchmarking field after a few things are addressed (see comments below).

> **[AR1-1] We thank the reviewer for the constructive feedback that helped us to improve the manuscript. We describe our approach to addressing the reviewer's specific comments below.**

**Major comments:**
[RC1-2] I have only one major comment, which is the need to include a result sub-section for limitations of the approach.

> **[AR1-2] We agree with the reviewer that this would be a valuable addition to the manuscript and included a paragraph detailing limitations of the approach in Section 3.3.3., lines 462 – 524 of the revised manuscript.**

Other comments (mostly minor).
Mostly suggestions for improving tables and general readability for the reader.
**Abstract:**
[RC1-3] Is very well written, but the spatial scale is defined almost at the end of the abstract. To aid the reader, a mention of your spatial scales is needed in line 11 (e.g. Previous comparisons "in North America"), line 14 (e.g. varies with spatial scale "(pixel, country, continent)"), line 19 (North American carbon balance "at any spatial scale").

> **[AR1-3] We updated the text to mention and define the spatial scales in line 12, 14 – 15, and 20.**

**Introduction:**
[RC1-4] Line 27.- change "beyond the plot scale" for "greater than a plot (1km2)"

> **[AR1-4] We incorporated this change on lines 29 – 30.**

[RC1-5] Line 28.- change "larger" for "broader"

**[AR1-5] We incorporated this change on line 31.**

[RC1-6] Second paragraph: incomplete ideas. At least two things need to be included here: 1) a line that helps the reader connecting the ideas with the next paragraph (i.e. "the issue arises from the particular characteristics of TBMs and Inversions") and 2) a stronger argument for the need to reduce uncertainties beyond an academic interest (e.g. reducing uncertainty also helps creating better mitigation policies).

**[AR1-6] We edited the introduction in accordance with this suggestion on lines 39 – 43.**

[RC1-7] Line 48.- actually the highest discrepancies across models are likely how they incorporate land use change, vegetation dynamics and fire (I believe far more than nitrogen or permafrost). Perhaps is worth mentioning a list (e.g. however other large discrepancies also arise from the different approaches on how to model land use change, vegetation dynamics and fire).

**[AR1-7] We added a sentence detailing the various potential sources of discrepancies across models in lines 51 – 53.**

[RC1-8] Line 48.- is not only differences in parametrization, but also on the driving data. TRENDY models sorted this issue by employing the same forcing and protocol, otherwise this is a major issue of variability. In summary, there are three elements that create model discrepancies: 1) structure, 2) parametrization and 3) driving data.

**[AR1-8] We agree with the reviewer's comment and edited this sentence on lines 53 – 55 of the revised manuscript.**

[RC1-9] Line 50.- you can take the argument further (since you are filtering models in this work), argue that you by addressing the sources of uncertainty you can benchmark model results, which can quickly lead to overall model improvement (e.g. if you know which model structure yields more realistic results, you can push other models to incorporate said structure).

**[AR1-9] We edited the manuscript to include the argument that this approach can help with model improvement by allowing for quick identification of which models yield more realistic results on lines 137 – 140.**

[RC1-10] Line 52.- the first sentence is too long. You could perhaps start with something simpler (e.g. "For inversions, uncertainties arise from several measurement and processing aspects").

**[AR1-10] We modified the first sentence as suggested on lines 61 – 62.**

[RC1-11] Line 57.- why? Can you explain a bit why we find large spread despite having high data availability?

**[AR1-11] While high data availability helps to reduce variability, it is not the only cause of variability in inverse modeling and it is possible that other sources of uncertainty are also significant contributors to variability. Variability can arise from other aspects of the inverse modeling framework such as choice of priors, model resolution, transport model, or fossil fuel inventory and there is not a clear consensus on the main source of uncertainty (Gaubert et al., 2019; Kondo et al., 2020). For example, some studies have found transport errors to be a primary source of uncertainty (Peylin et al. 2011; Schuh et al., 2019) while other studies have shown that fossil fuel emission uncertainties play a key role in variability (Gurney et al. 2005; Peylin et al. 2011;  Saeki and Patra 2017). We discuss this point on lines 66 – 71 of the revised manuscript.**

[RC1-12] Line 58.- change "Another challenge" for "Despite their usefulness, a key limitation"

**[AR1-12] We changed "Another challenge" to "Another limitation" on line 71 to reflect the reviewer's suggested term.**

[RC1-13] Line 62.- change "how well we understand" for "our understanding of"

**[AR1-13] We incorporated this change on line 76.**

[RC1-14] Line 64.- is not that the confidence in both types of model increases, I would say that they become more reliable as sources of "realistic" information.

**[AR1-14] We agree and edited the wording to reflect this on lines 77 – 78.**

[RC1-15] Lines 66- 70. Please expand on the examples, as you are comparing different spatial scales and regions. Particularly, please provide detailed examples for previous results in North America.

**[AR1-15] We expanded this paragraph to include concrete examples of agreement/disagreement between bottom-up and top-down approaches with a focus on providing examples specific to North America on lines 83 – 88.**

[RC1-16] Line 70 (sixth paragraph). The paragraph provides incomplete reasoning and needs to be improved grammatically. You could start with an opening line such as: "There are different approaches to compare TBMs with inversions. On the first hand, there are direct comparisons of the means, which is usually referred to as "agreement" (citation); on the other hand, there are approaches centered on the variability, which we defined as "consistency" (citation). The first provides XXX type of information, while the second XXX".

**[AR1-16] We updated the paragraph to improve the grammar and provide clarity on the definitions of consistency and agreement on lines 90 – 95.**

[RC1-17] Line 75.- add "Previous" to the beginning of the sentence. Change "reveal" to "have revealed". Add- the agreement "between estimates".

**[AR1-17] We modified the sentence according to the feedback given on line 97.**

[RC1-18] Line 76.- remove "do"

**[AR1-18] We incorporated this change on line 98.**

[RC1-19] Line 77.- move ",however," to the beginning of the sentence.

**[AR1-19] We incorporated this change on line 99.**

[RC1-20] Line 78.- remove the "," before the "and".

**[AR1-20] We incorporated this change on line 100.**

[RC1-21] Line 87. Opening sentence is too long. Perhaps start with: "A key step forward is to look at agreement and comparison across scales".

**[AR1-21] We updated the sentence to be shorter on line 111.**

[RC1-22] Line 101.- This argument is always complicated. In theory, you would expect that models who provide better estimates in historical runs, would be better suit for future projections; however, this is not always the case. For example, models that include a N cycle usually perform worst than C-only models in present-day conditions, however they are likely better at recreating future scenarios where N becomes limiting for NPP.

I believe you can leave the sentence as is in the introduction, but the arguments presented need to be included in the discussion.

> **[AR1-22] We agree that better performance under present conditions does not necessarily translate to better performance under future conditions. We now allude to this in the introduction (lines 137 – 140) and discussion of the revised manuscript (lines 522 – 524).**

[RC1-23] Lines 101-103. Repetitive, you have already defined the terms. Not needed.

> **[AR1-23] We removed the definitions of the terms to avoid repetition in lines 125 – 126.**

[RC1-24] Line 116. Change "North American NEE" for "NEE in North America"

> **[AR1-24] We incorporated this change in line 142.**

**Methods**

[RC1-25] One aspect that is complicated is how very little information there is to benchmark models based on seasonality. There are only 4 towers employed in the seasonality analysis, all of which are located in croplands; even in the larger Tower compendium, there is little representation across drylands which have been shown to drive most of the IAV of the CO2 cycle. This needs to be addressed in the discussion into detail.

> **[AR1-25] We agree that just because models perform well for these four towers does not necessarily mean the models will perform well elsewhere. We note that Sun et al. (2021) found that TBMs that showed strong carbon uptake in croplands during the growing season were in better agreement with atmospheric observations from 44 towers, highlighting that croplands are key in capturing North American carbon fluxes. We also note that the tower location within a specific biome does not necessarily mean its observations are only influenced by fluxes within that biome. We now reference Sun et al., (2023) in lines 463 – 465 of the revised manuscript as they performed an analysis demonstrating this (Extended data Fig. 2).**

> **However, this comment prompted us to do further analysis into the impact of including additional towers on our results, which we included in the revised version of the manuscript in lines 463 – 472. We found that including more towers in other parts of North America (as a result of loosening the criteria for inclusion in the seasonality analysis) did not change our overall conclusions.**

**Specifically, we kept the requirement that a tower must have observations for at least half of the days in the study period, but we loosened the requirement of maximum allowable data gap from 31 to 80 consecutive days. This resulted in four additional towers being included (SGP, AMT, OFR, and EGB). Using these towers we first re-calculated the seasonality metric using only these four towers (SGP, AMT, OFR, and EGB) and then re-calculated the seasonality metric using these towers in addition to the towers that meet the original selection criteria (AME, WKT, ETL, LEF, SGP, OFR, AMT, and EGB).**

**We found that ten models perform well based on the seasonality metric regardless of which subset of towers is used, while six and four additional models also meet the metric when original and new sets of four towers are used, respectively. While this impacts the consistency of ensembles based on the seasonality metric alone, the impact on the consistency and agreement of models that meet all three metrics is minimal. We included a discussion of this additional analysis in the revised manuscript in lines 463 – 472 and included a figure showing this (Fig. S8).**

[RC1-26] Lines 147-153. One key issue with TRENDY data is the land-mask employed to remove ocean fractions. This needs to be clearly stated. If the data was crop first at the original resolution (0.5°) then regridded, this is not an issue; but if you regridded first (to a 1x1 grid), the estimates for NEE become much larger. Please specify how you perform post-processing of the data.

**[AR1-26] The native resolution of the TRENDY models is not uniform across models. For this reason, we first regridded the data to 1°x1° resolution using xESMF conservative regridding algorithm (Zhuang et al., 2023, Zenodo) for the global data and then cropped the data to a uniform North American domain. The conservative regridding method preserves the source field's integral (e.g., total fluxes for North America), ensuring that the total NEE at the native resolution is preserved after regridding to 1°x1° resolution. As these models are all terrestrial biosphere models, the ocean fluxes are not represented and therefore cannot explain the gap between bottom-up and top-down NEE estimates. We clarify the post-processing steps used in lines 190 – 191 of the revised manuscript.**

[RC1-27] Line 175. Why not using MODIS GPP?

**[AR1-27] We chose to use APAR because we wanted to use the simplest data-driven baseline possible. We view APAR as a simpler and more direct baseline than MODIS GPP because MODIS GPP is modeled using multiple parameters and is therefore itself a type of model (Running and Zhao, 2015). We added in an**

**explanation for why we did not use MODIS GPP in the revised manuscript on lines 223 – 226.**

**Acknowledgements**

[RC1-28] Please notice that the TRENDY data policy states that you need to clearly acknowledge them for using their data.

**[AR1-28] We did confirm with the data provider that the acknowledgement as written is appropriate before we submitted the original manuscript.**

**Results**

[RC1-29] Line 276.- Please change the "however" to the beginning of the paragraph, and merge this paragraph with the previous one.

**[AR1-29] We moved the "however" to the beginning of the paragraph as suggested on line 330, though we feel that keeping these paragraphs separate helps to convey the two separate ideas within them.**

[RC1-30] Line 371-379.- I believe these values should be presented as a table. Particularly show the how the mean and deviation for the region (and land categories) changes with model filtering.

**[AR1-30] We created a new table that includes these values (see Table 2).**

[RC1-31] Please add a section on limitations of the study and the approach. I believe this should be clearer. While the results are really interesting and promising, several data-limitation issues are presented (mentioned above).

**[AR1-31]. Please refer to response [AR1-2].**

**Tables.**

[RC1-32] Table 1.- please organize the table by type of ensemble instead of model name.

**[AR1-32] We organized the table by type of ensemble instead of model name (see Table 1).**

[RC1-33] Tables 1 & 2.- I strongly suggest to merge both tables. A simple solution is to add four columns to table one (one for each metric and the total). This way the reader can quickly see which models meet which metrics.

**[AR1-33] We merged the two tables into one table (see Table 1).**

**Figures**
[RC1-34] Figure 2.- I believe this figure should go into the supplementary

> **[AR1-34] We believe this figure is helpful in illustrating the approach we use for determining consistency across spatial scales and therefore think it is useful to have this figure in the main text.**

[RC1-35] Figure 5.- Why is there no comparison for grasslands and drylands? They represent a major proportion of land across NA!

> **[AR1-35] This comes down to the lack of data available from ObsPack in these biomes so we only compared the biomes with the largest data constraint. We reference Sun et al., (2023), who did an analysis showing the data constraint for individual biomes in lines 437 – 439.**

**References used in the response to Reviewer # 1**
Gaubert, B., Stephens, B. B., Basu, S., Chevallier, F., Deng, F., Kort, E. A., Patra, P. K., Peters, W., Rödenbeck, C., Saeki, T., Schimel, D., Van der Laan-Luijkx, I., Wofsy, S., and Yin, Y.: Global atmospheric CO2 inverse models converging on neutral tropical land exchange, but disagreeing on fossil fuel and atmospheric growth rate, Biogeosciences, 16, 117–134, https://doi.org/10.5194/bg-16-117-2019, 2019.

Gurney, K. R., Chen, Y.-H., Maki, T., Kawa, S. R., Andrews, A., and Zhu, Z.: Sensitivity of atmospheric $CO_2$ inversions to seasonal and interannual variations in fossil fuel emissions, J. Geophys. Res., 110, D10308, https://doi.org/10.1029/2004JD005373, 2005.

Jiawei Zhuang, raphael dussin, David Huard, Pascal Bourgault, Anderson Banihirwe, Stephane Raynaud, Brewster Malevich, Martin Schupfner, Filipe, Sam Levang, André Jüling, Mattia Almansi, RichardScottOZ, RondeauG, Stephan Rasp, Trevor James Smith, Jemma Stachelek, Matthew Plough, Pierre, … Xianxiang Li. (2023). pangeo-data/xESMF: v0.7.1 (v0.7.1). Zenodo. https://doi.org/10.5281/zenodo.7800141

Kondo, M., Patra, P. K., Sitch, S., Friedlingstein, P., Poulter, B., Chevallier, F., ... & Ziehn, T. (2020). State of the science in reconciling top-down and bottom-up approaches for terrestrial CO2 budget. Global change biology, 26(3), 1068-1084.

Peylin, P., Houweling, S., Krol, M. C., Karstens, U., Rödenbeck, C., Geels, C., Vermeulen, A., Badawy, B., Aulagnier, C., Pregger, T., Delage, F., Pieterse, G., Ciais, P., and Heimann, M.: Importance of fossil fuel emission uncertainties over Europe for $CO_2$ modeling: model intercomparison, Atmos. Chem. Phys., 11, 6607–6622, https://doi.org/10.5194/acp-11-6607-2011, 2011

Running, S. W., & Zhao, M. (2015). Daily GPP and annual NPP (MOD17A2/A3) products NASA Earth Observing System MODIS land algorithm. *MOD17 User's Guide*, *2015*, 1-28.

Saeki, T. and Patra, P. K.: Implications of overestimated anthropogenic $CO_2$ emissions on East Asian and global land $CO_2$ flux inversion, Geoscience Letters, 4, 9, https://doi.org/10.1186/s40562-017-0074-7, 2017.

Schuh, A. E., Jacobson, A. R., Basu, S., Weir, B., Baker, D., Bowman, K., Chevallier, F., Crowell, S., Davis, K. J., Deng, F., Denning, S., Feng, L., Jones, D., Liu, J., and Palmer, P. I.: Quantifying the Impact of Atmospheric Transport Uncertainty on CO2 Surface Flux Estimates, Global Biogeochem. Cy., 33, 484–500, https://doi.org/10.1029/2018GB006086, 2019.

Sun, Wu, Yuanyuan Fang, Xiangzhong Luo, Yoichi P. Shiga, Yao Zhang, Arlyn E. Andrews, Kirk W. Thoning, Joshua B. Fisher, Trevor F. Keenan, and Anna M. Michalak. "Midwest US croplands determine model divergence in North American carbon fluxes." AGU Advances 2, no. 2 (2021): e2020AV000310.

Sun, W., Luo, X., Fang, Y., Shiga, Y. P., Zhang, Y., Fisher, J. B., Keenan, T. F., and Michalak, A. M.: Biome-scale temperature sensitivity of ecosystem respiration revealed by atmospheric CO2 observations, Nat Ecol Evol, 7, 1199–1210, https://doi.org/10.1038/s41559-023-02093-x, 2023.

Reviewer # 2 (Remarks to the Author):
We thank the reviewer for their comments and suggestions. Please find our response below in **bold**.

[RC2-1] The manuscript is generally well written and delves into the discrepancies between NEE as estimated by terrestrial biosphere models (TBMs) and those derived from top-down atmospheric inversions across different scales. Addressing this topic is important in developing reliable carbon budgets. However, the manuscript misinterprets the NEEs estimated by these TBMs. Furthermore, the discussion section seems to lack comprehensive analysis, leaving out essential arguments that could better support the authors' conclusions. These sections would benefit significantly from a detailed review and subsequent refinement.

> **[AR2-1] We thank the reviewer for the constructive comments that helped to improve the manuscript. We address the reviewer's specific comments in the responses below.**

**Comments:**
[RC2-2] 1) In Table 1, results from both the BG1 and SG3 scenarios of MsTMIP are presented. However, in the methods section, only the usage of BG1 results is detailed. Could you please provide an explanation for this discrepancy?

**[AR2-2] In lines 138 - 140 we state that we use both BG1 and SG3 simulations. We updated the wording to more clearly convey that we use both scenarios in lines 161 – 163.**

[RC2-3] 2) In the MsTMIP project, the participating TBMs do not estimate the NEE directly; instead, they utilize the stock change approach. It's important to note that while some models incorporate factors like fire and harvesting into their simulations, others do not. This distinction should be clearly addressed in the manuscript. Additionally, while certain models use fire and harvesting data in calculating the NEE, Line 175 of the manuscript only acknowledges the photosynthesis and ecosystem respiration factors. This discrepancy needs addressing.

**[AR2-3] We appreciate this feedback from the reviewer, which led us not only to edit the description in the manuscript, but to explore more deeply how differences in the processes incorporated in the various models included in the MsTMIP ensemble impact the consistency across the ensemble. We included the results of this analysis in the revised version of this manuscript in lines 354 – 365 and lines 503 – 512.**

**More specifically, we examined how using a simple definition of NEE (respiration minus GPP) impacts both consistency within the MsTMIP ensemble and the agreement between the MsTMIP models and inversions. Surprisingly, we found that estimates from the MsTMIP ensemble are less consistent across models when using only GPP, heterotrophic respiration and autotrophic respiration in the calculation of NEE. In other words, the models are more consistent when they include other components of NEE, even though the included components differ from model to model. This seems to suggest that models may implicitly target a presumed net land sink irrespective of the processes included. Only four models define NEE as NEE = $R_h$ + $R_a$ + $F_{disturbance}$ + $F_{product}$ - GPP (CLM4, CLM4VIC, TEM and VEGAS) and how well these models agree with inversions varies by biome. We did find that agreement with inversions was slightly better when a simple definition of NEE is used. We included a discussion of this new analysis in lines 354 – 365 and lines 5034 – 512 and included an associated figure in the revised manuscript (Fig. S7).**

[RC2-4] 3) In your study, I noticed that a singular value was presented for all the TBMs and AIMs. Could you clarify how you combined the results from these models on Line 150? If you simply averaged them, I suggest referencing the 'integration approach' detailed by Schwalm et al. (2015), Toward 'optimal' integration of terrestrial biosphere models, Geophysical Research Letters. Given the significance of spatial patterns in this study, relying solely on a simple average might introduce notable uncertainties. It's worth

mentioning that the NEEs estimated by different TBMs can vary significantly. Hence, it would be beneficial to analyze each model's estimates separately and delve into the biogeochemical processes that might account for the discrepancies observed between top-down and bottom-up approaches.

> **[AR2-4] We clarified that we use the model ensemble average for TBMs and inversions in lines 193 – 196. In the context of Schwalm et al., (2015), it is noteworthy that one of the key findings was that the added complexity of skill-based integration does not materially change flux estimates based on TBM ensembles . Given this, we feel that the model ensemble average approach used here is appropriate. We agree that it is important to mention the different ways that NEE is estimated and we discuss this in the methods section 2.1.1(lines 163 – 172) and results section (lines 354 – 365 and lines 503 – 512) of the revised manuscript. We agree that it is important to discuss potential reasons for the discrepancies between bottom-up and top-down approaches. We therefore include a discussion of the sensitivity analyses we performed to determine potential causes of discrepancies and reference papers that have also looked into this in Section 3.3.3, lines 463 – 512. We believe that analyzing each model's estimates separately and determining which biogeochemical processes might account for the observed discrepancies between bottom-up and top-down approaches is beyond the scope of this study, however. One of the main goals of this study is to identify which models perform better in terms of reproducing basic features of observed atmospheric CO2 variability so that further analysis can be done to understand the potential causes in specific models.**

[RC2-5] 4) To explain the discrepancy between top-down and bottom-up estimates of NEE, numerous studies have been conducted, including notable publications by Peter A. Raymond and David E. Butman. The discrepancy is attributed to the lateral carbon flux of dissolved organic carbon, particulate organic carbon, and carbon in inorganic formats. Given that these TBMs have been calibrated and validated using field measurements, such as soil organic carbon, it is possible to incorporate the lateral carbon fluxes when estimating each carbon pool. Unfortunately, this process seems to have been overlooked in the current discussion.

> **[AR2-5] We agree that it is important to discuss the role of lateral fluxes and thank the reviewer for this suggestion. We added a paragraph to discuss the impact of lateral fluxes on the discrepancy between bottom-up and top-down estimates in lines 484 – 501. This comment also prompted us to explore the role of lateral fluxes by comparing the impact of including two different lateral**

flux estimates from Byrne et al. (2023) on agreement between bottom-up and top-down methods.

More specifically, we added the gridded lateral fluxes from river export, crop trade, and wood trade to TBMs to make them more comparable with fluxes seen by inversions and we assess the impact of doing so. In our analysis, we use two different estimates of river export from Byrne et al., (2023). The first is the gridded product which incorporates data from Global NEWS and the second is the same gridded product rescaled so that the total river export from the gridded product equals the reported country total river export in Byrne et al., (2023), which is the mean of two models (Global NEWS and DLEM). It is important to note the differences in these two river export estimates highlight some of the uncertainties associated with estimating lateral fluxes and that there are significant uncertainties associated with estimating river export (Byrne et al., 2023; Drake et al., 2017). Given this, we are not fully able to account for lateral fluxes, but rather show the potential impact that lateral fluxes could have.

We find that at the North American scale, incorporating lateral fluxes improved agreement between bottom-up and top-down models somewhat, but the change was not sufficient to explain discrepancies. For example, in the deciduous broadleaf & mixed forests and cropland biomes, lateral fluxes only partially explained discrepancies and applying the seasonality, variability, and magnitude metrics still has an impact on improving agreement between bottom-up and top-down estimates (see Fig. S9 in the revised manuscript). This is in contrast to the evergreen and needleleaf forest biome where the inclusion of lateral fluxes led to better agreement between inversions and TBMs for the subset with all models included, but ultimately exacerbated differences between bottom-up and top-down methods once models that meet all three criteria were selected.

[RC2-6] Line 75: The lateral carbon flux should be discussed see Casas-Ruiz et al. (2023), Integrating terrestrial and aquatic ecosystems to constrain estimates of land-atmosphere carbon exchange, Nature Communications.

**[AR2-6] We added in a discussion of how lateral carbon fluxes impact agreement between bottom-up and top-down models, as discussed in [AR2-5].**

[RC2-7] Line 101: To adequately convey the concept of "consistency" in the article, it's crucial to delve into the differences among the models. These models vary significantly in their simulation processes, leading to considerable variations in NEE. It's important to note that the models participating in the MsTMIP project use the same input data. I am not

familiar with the TRENDY project. Ensure that both projects utilize the same input data; if not, the differing input data could be a significant source of variance.

**[AR2-7] Please refer to [AR2-3].**

[RC2-8] Line 204: From where did you obtain the biome map? Additionally, did both MsTMIP and TRENDY projects utilize the same biome map?

**[AR2-8] We added a sentence detailing where the biome map comes from and whether MsTMIP-v2 and TRENDY-v9 use the same biome map in lines 177 – 187. We used the same biome map as Shiga et al., 2018 and Sun et al., 2021 that is based on an IGBP landcover classification map. MsTMIP-v2 imposes a consistent biome map for all models (Wei et al., 2014), while models from TRENDY use various sets of plant functional types (PFTs), resulting in differences in land cover representations, though they do use common land use and land cover change (LULCC) forcing data (Seiler et al., 2022). Understanding the impact of various biome maps used by models is difficult as few models provide outputs at the resolution necessary to do a comprehensive analysis such as evaluating whether specific PFTs are present at in situ observation sites (Seiler et al., 2022). However, Sun et al., (2021) did compare the impact of model-specific biome classifications for four models that provided PFT information at finer resolutions and showed that model-specific biome classifications did not impact their overall conclusions.**

[RC2-9] Line 311: Could you please list the two-thirds of TBMs and analyze the potential reasons for their behavior? Specifically, do these models incorporate certain key processes?

**[AR2-9] Figure S2 shows which models represent the space-time variability of atmospheric $CO_2$ less well than APAR. We now reference this figure line 371. We believe that looking into individual models is beyond the scope of this study, but in [AR2-3] we describe how we have explored the impact of what processes are included on consistency and agreement.**

[RC2-10] Line 351: TBMS – TBMs

**[AR2-10] We incorporated this change on line 412.**

[RC2-11] Line 380: The current discussion is insufficient. It's essential to address the role of lateral carbon flux in causing discrepancies between bottom-up and top-down estimates. Furthermore, the method of NEE calculation across different models should be discussed.

NEE estimation approaches for these TBMs:

NEE=-GPP+TR

NEE=-GPP+TR+Fire

NEE=-GPP+TR+ Harvesting

NEE=-GPP+TR+Fire+Harvesting

**[AR2-11] Please refer to [AR2-3] where we discuss how we will explore the role of which processes are included in models on consistency and agreement, and please refer to [AR2-5] where we describe how we will address the role of lateral fluxes.**

References used in the response to Reviewer # 2
Byrne, B., Baker, D. F., Basu, S., Bertolacci, M., Bowman, K. W., Carroll, D., ... & Zeng, N. (2023). National CO 2 budgets (2015–2020) inferred from atmospheric CO 2 observations in support of the global stocktake. *Earth System Science Data*, *15*(2), 963-1004.

Drake, T. W., Raymond, P. A., & Spencer, R. G. (2018). Terrestrial carbon inputs to inland waters: A current synthesis of estimates and uncertainty. *Limnology and Oceanography Letters*, *3*(3), 132-142.

Schwalm, C. R., Huntzinger, D. N., Fisher, J. B., Michalak, A. M., Bowman, K., Ciais, P., ... & Zeng, N. (2015). Toward "optimal" integration of terrestrial biosphere models. *Geophysical Research Letters*, *42*(11), 4418-4428.

Seiler, C., Melton, J. R., Arora, V. K., Sitch, S., Friedlingstein, P., Anthoni, P., ... & Zaehle, S. (2022). Are terrestrial biosphere models fit for simulating the global land carbon sink?. *Journal of Advances in Modeling Earth Systems*, *14*(5), e2021MS002946.

Shiga, Y. P., Tadić, J. M., Qiu, X., Yadav, V., Andrews, A. E., Berry, J. A., & Michalak, A. M. (2018). Atmospheric CO2 observations reveal strong correlation between regional net biospheric carbon uptake and solar-induced chlorophyll fluorescence. *Geophysical Research Letters*, *45*(2), 1122-1132.

Sun, Wu, Yuanyuan Fang, Xiangzhong Luo, Yoichi P. Shiga, Yao Zhang, Arlyn E. Andrews, Kirk W. Thoning, Joshua B. Fisher, Trevor F. Keenan, and Anna M. Michalak. "Midwest US croplands determine model divergence in North American carbon fluxes." AGU Advances 2, no. 2 (2021): e2020AV000310.